# Integration of mathematical modeling and economics approaches to evaluate strategies for control of *Salmonella* Dublin in a heifer-raising operation

**Sebastian Llanos-Soto**[1]*, **Martin Wiedmann**[2], **Aaron Adalja**[3], **Christopher Henry**[1], **Paolo Moroni**[1,4], **Elisha Frye**[1], **Francisco A. Leal Yepes**[1], **Renata Ivanek**[1]

1 Department of Population Medicine and Diagnostic Sciences, Cornell University, Ithaca, New York, United States of America, 2 Department of Food Science, Cornell University, Ithaca, New York, United States of America, 3 School of Hotel Administration, Cornell University, Ithaca, New York, United States of America, 4 Dipartimento di Medicina Veterinaria e Scienze Animali, Università degli Studi di Milano, Lodi, Italy

* sgl67@cornell.edu

## Abstract

*Salmonella* Dublin infections in heifer-raising operations (HROs) cause animal health and economic losses for these operations and represent a pathogen source for dairy farms obtaining replacement heifers from HROs. To improve control of *S.* Dublin, we (i) developed a mathematical model of *S.* Dublin transmission on a HRO, (ii) evaluated the vaccine effectiveness and cleaning improvements for controlling the infection, and (iii) evaluated the influence of infection and control strategies on the HRO's operating income. We developed a modified Susceptible-Infected-Recovered-Susceptible model of *S.* Dublin spread in a batch-stocking HRO post-introduction of an index case, with stochasticity introduced through Monte Carlo simulations. Epidemiological outcomes (*S.* Dublin-induced deaths and abortions during raising and *S.* Dublin carriers and asymptomatic infections among raised replacement heifers) and operating income per 100-head raised on a HRO over a 2-year simulation were compared between control scenarios. We validated our model against *S.* Dublin infection data in cattle. Partial rank correlation coefficient analysis and classification trees were used to determine parameter influence on model outcomes. Our model predicts a median of 37 carriers and 92 asymptomatic infections among raised replacement heifers out of 2,330 heifers that departed the operation by the end of the 2-year simulation period, suggesting a relevant role of HROs in spreading *S.* Dublin. Increasing barn floor cleaning frequency (to a maximum of 12x per day) meaningfully reduced the *S.* Dublin epidemiological outcomes and improved the HRO's operating income. Depending on the cost of cleaning, the median operating income increased between 1.2% to 10.6% in the first year when cleaning 12x per day compared to baseline (cleaning 1x per week). In most cost

**Data availability statement:** Information used for model validation is described in the S1 Appendix related to this article. The code for the model is available online at: https://github.com/IvanekLab/SDublin.git.

**Funding:** This study was financially supported by the National Institute of Food and Agriculture (https://www.nifa.usda.gov/) in the form of grants (7000433, 1016738 and 7005699) received by RI. This study was also financially supported by the Cornell Institute for Digital Agriculture (CIDA) (https://digitalagriculture.cornell.edu/research-support), Cornell University, in the form of a Research Innovation Fund (RIF) award received by SL-S.

**Competing interests:** The authors have declared that no competing interests exist.

scenarios, predictions do not support using a vaccine that solely reduces mortality, even when paired with stringent cleaning measures. The developed model is expected to aid efforts to control *S.* Dublin in HROs.

## Introduction

Cattle movement between heifer-raising operations (HROs) and dairy farms increases exposure opportunities and susceptibility to *Salmonella* infection due to stress associated with the transition between the two environments, dietary changes, and interaction with new animals [1]. Among *Salmonella* serotypes, *Salmonella* Dublin represents a major threat to HROs due to its ability to cause a long-term infection in calves post-recovery from clinical disease (i.e., carrier state; [2]), thus providing an opportunity for the infection to be introduced into the HRO (and eventually spread to other herds) unnoticedly. Unlike other non-typhoidal *Salmonella* serotypes, *S.* Dublin is associated with more severe invasive disease in humans and host-adapted to cattle, with the ability to persist within herds for extended periods [3]. The establishment of *S.* Dublin carriers within cattle herds hinders its control and eradication due to its ability to persist asymptomatically in infected cattle, periodically trigger outbreaks of disease, and evade detection due to intermittent shedding [4,5]. The importance of HROs in pathogen dissemination among dairy operations is particularly concerning, considering the widespread multi-drug resistance (MDR) among *S.* Dublin isolates in the US [3,6,7]. The MDR in *S.* Dublin leads to significant challenges in veterinary treatment and control [8] and serves as a source of resistance genes for other *Salmonella* serotypes and bacterial pathogens affecting dairy cattle [9]. Because *S.* Dublin can spread from dairy farms to HROs via calves and from HROs to dairy farms via pregnant replacement heifers, HROs are positioned as potential amplifiers and disseminators of this pathogen within the dairy farming system. Once introduced into dairy herds, *S.* Dublin can cause clinical disease in calves and abortion in cows and poses a risk to human health through the potential for bloodstream infections following the consumption of contaminated animal products such as undercooked beef or raw milk [3,10]. To diminish the occurrence of infection in dairy herds, including MDR strains, it is critical to understand *S.* Dublin infection dynamics in HROs and determine the most cost-effective approach to reducing transmission of *S.* Dublin in HROs.

The primary route of *S.* Dublin infection in cattle is oral ingestion of the bacterium, with fecal contamination serving as a key contributor to the contamination of feed, water, pen surfaces, bedding, and equipment [11]. In addition to feces, *S.* Dublin is shed in bodily fluids such as milk, saliva, and nasal secretions [5]. Asymptomatic carriers can intermittently shed the pathogen, leading to ongoing environmental contamination and continued exposure of susceptible animals to infectious sources [2,5]. Notably, *S.* Dublin can survive for extended periods in the environment, including in bovine feces slurry, where it has been reported to persist for months [12], enabling sustained transmission even in the absence of clinically ill animals. Because the contaminated environment plays a key role in *S.* Dublin transmission and persistence

within cattle operations, incorporating environmental factors is critical for accurately characterizing its epidemiology and informing effective control strategies.

The ability of *S*. Dublin to cause severe systemic disease in young cattle, its widespread MDR, and its ability to persist long-term undetected within herds warrant its assessment separately from other *Salmonella* serovars. By synthesizing previously collected data and information published in the literature, mathematical models can provide valuable insights into *S*. Dublin transmission dynamics and assist in identifying knowledge gaps regarding its microbiology and epidemiology in HROs. For instance, previous modeling studies have explored the importance of environmental contamination in the persistence of *Salmonella* spp. (including *S*. Dublin) in cattle [13–16] and the role of clinically infected, long-term infected, and supershedder individuals in *S*. Dublin transmission dynamics [16,17]. Mathematical models have also been used to assess *what-if* scenarios and determine the importance of mitigation and biosecurity measures to prevent *S*. Dublin introduction and dissemination in dairy herds [13,16–21]. Importantly, mathematical models can account for the variability in real-world processes and the uncertainty in our understanding of those processes by incorporating stochasticity in parameter values, thus providing more realistic and comprehensive predictions of disease dynamics [22]. Predictions from mathematical models can be used to understand the economic impacts of infectious disease and to identify the most financially appropriate mitigation strategy through economic analyses [23]. Findings obtained through the combined application of epidemiological and economic approaches are expected to assist farmers and veterinarians in controlling *S*. Dublin infections in HROs. To date, available modeling studies have provided valuable insights into understanding *Salmonella* epidemiology in dairy operations and its economic impacts. However, these studies have not examined the epidemiology of *S*. Dublin in HROs, nor have they applied an economic modeling approach to evaluate cost-effective mitigation strategies in the US context.

To improve control of *S*. Dublin on HROs, in this study we combined mathematical modeling and economic analysis to (i) develop a mechanistic compartmental mathematical model of *S*. Dublin transmission on a HRO, (ii) evaluate the effectiveness of a commercial vaccine and different cleaning strategies for controlling *S*. Dublin infection, and (iii) evaluate the influence of infection and control strategies on the HRO's operating income.

## Materials and methods

### Heifer-raising operation characteristics

In this study, the model system is a US HRO that raises replacement heifers after weaning. Calves enter the HRO at 90 days old and depart at 630 days of age (i.e., for a total raising period of 540 days), preceding their first calving (note that pre-weaned calves and calved heifers are not considered in the model). Heifers' arrival and departure age were determined based on data from the USDA [24]. The model HRO represents the system present in 53.9% of HROs in the USA in 2011 (Fig 1, [24]); the remaining 46.1% (not represented by the model) consist of operations that manage animals entering and exiting at different stages of the raising period. The HRO model developed here considers cattle groups in three age categories in the barn, each one with a designated subindex to help describe model parameters: (i) weaned calves ($_c$), (ii) growing heifers ($_g$), and (iii) pregnant heifers ($_p$). The model was implemented as ordinary differential equations (ODE) in R v. 4.2.0 [25] using the *deSolve* R package [26]. The model is comprised of epidemiological and economic modules.

### Epidemiological module

The epidemiological module is based on a modified Susceptible-Infected-Recovered-Susceptible (SIRS) compartmental model (Fig 2; model parameters are defined in Table 1). This module describes *S*. Dublin spread inside a free-stall barn in a HRO after introducing an asymptomatic infectious weaned calf (index case) at time $t = 0$. The time unit of the ODE model is a day. This model considers a density-dependent transmission process, where the risk of infection increases with the density of bacteria in the environment, which increases with cattle density on the farm. We chose a density-dependent

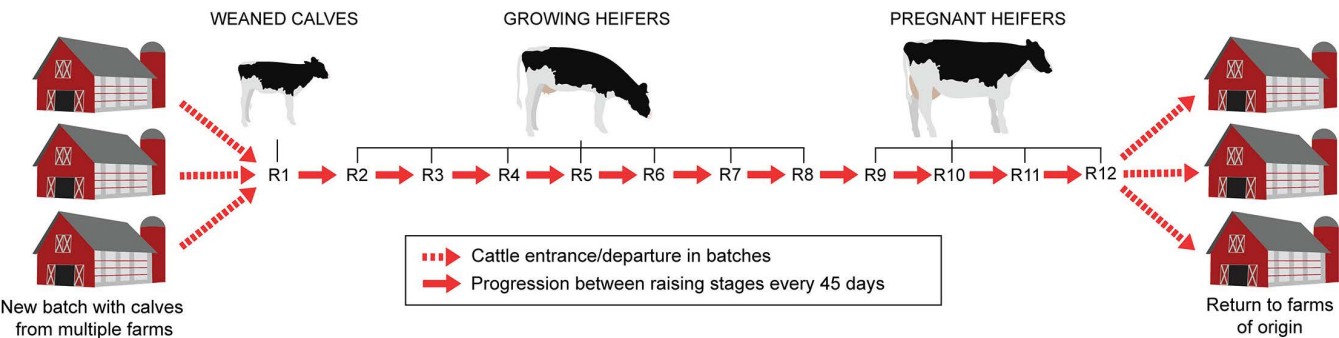

**Fig 1. Heifer-raising operation (HRO) workflow represented in the model.** Calves enter the operation being 90 days old (weaned) and return to their facility of origin after 540 days (630 days of age and pregnant). During this period, individuals progress through twelve raising stages: one as a weaned calf (R1; 90–135 days old), seven as growing heifers (R2 to R8; 136–450 days old), and four as pregnant heifers (R9 to R12; 451–630 days old). In contrast to other raising stages, pregnant heifers (R9 to R12) cannot die from infection, but they may experience abortion. Individuals transition between stages every 45 days to reflect the arrival of heifers into the HRO in batches and to allow for more precise estimation of age-specific costs, including losses due to death or abortion within each stage.

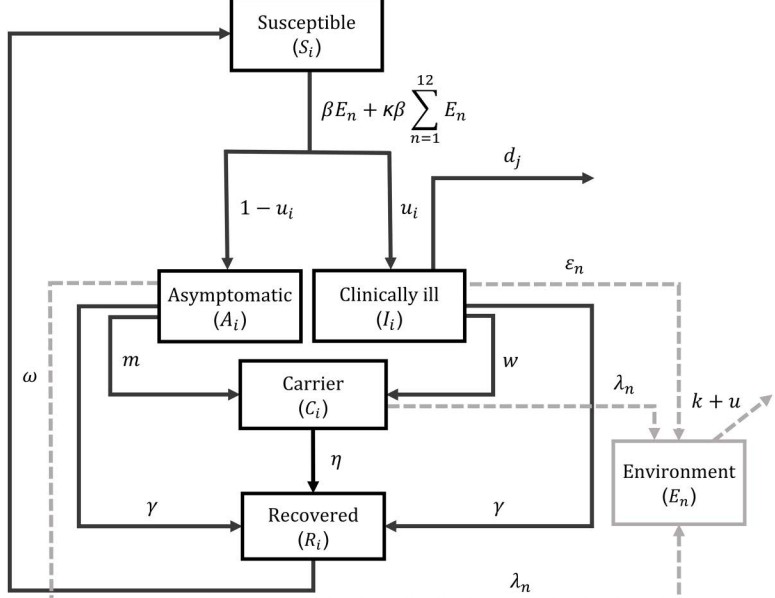

**Fig 2. Transmission dynamics of _Salmonella_ Dublin in a heifer-raising operation (HRO).** Transmission dynamics model of _S_. Dublin infection in a HRO in a baseline scenario in which no mitigation strategy has been implemented. Solid arrows represent an individual animal's movement among compartments and dashed arrows indicate _S_. Dublin shedding and the dynamics in the environment. A subscript _i_ indicates that a parameter value changes according to age category; while a subscript _j_ was added in parameter _d_ to indicate that only weaned and growing heifers can die from _S_. Dublin infection. Subscript _n_ indicates that the parameter varies in value according to the raising stage (R1 to R12). For compartment $E_n$, subscript _n_ represents the pen environment for each raising stage.

formulation to reflect the biological assumption that in a denser population, cattle are more likely to come into indirect contact through contaminated environment and spread the infection. Direct transmission was considered comparatively much less important in the spread of _S_. Dublin and was thus ignored. That is because direct transmission mainly occurs when a susceptible animal ingests fecal material from the recto-anal region of an infectious individual, while contamination of

**Table 1. Details about the epidemiological module parameters, including equations and values. Parameter notations, definitions, and values for *Salmonella* Dublin infection in a heifer-raising operation in a baseline model (without control strategies).**

| Notation | Definition (unit) | Equation[a] | Value/Mean (5th, 95th percentile) | Reference |
|---|---|---|---|---|
| $N$ | Number of cattle on the farm (individual) | N/A[b] | 1,000 | N/A |
| $\beta$ | Rate of environment-to-host transmission within a pen (*Salmonella* cell$^{-1}$ individual$^{-1}$ day$^{-1}$) | N/A | $10^{-10.0}$ | Assumed |
| $\kappa$ | Reduction coefficient for *S.* Dublin cross-pen transmission (dimensionless) | N/A | 0.005 | [27] |
| $D_{All}$ | Duration of stay in the *A* or *I* compartments (days) | Beta-PERT(3;17;65) | 22.0 (7.7–42.5) | [14,28] |
| $D_C$ | Duration of stay in the *C* compartment (days) | Beta-PERT(240;365;1,095) | 469.6 (271.5–728.3) | [2,29,30] |
| $D_R$ | Duration of stay in the *R* compartment (days) | Truncated Exponential(140;1/140)[c] | 239.3 (148.8–579.4) | [16,31] |
| $\gamma$ | Recovery rate from the *A* or *I* compartment (day$^{-1}$) | $1/D_{All}$ | 0.7 (0.02–1.2) | N/A |
| $\eta$ | Recovery rate from the *C* compartment (day$^{-1}$) | $1/D_C$ | 0.002 (0.001–0.004) | N/A |
| $\omega$ | Immunity loss rate (day$^{-1}$) | $1/D_R$ | 0.004 (0.002–0.007) | N/A |
| $u_w$ | Probability that a newly infected weaned heifer will develop clinical disease (enter the *I* compartment rather than the *A* compartment) (dimensionless) | Beta-PERT(0.05;0.25;0.5) | 0.3 (0.1–0.4) | [16,32] |
| $u_g$ | Probability that a newly infected growing heifer will develop clinical disease (enter the *I* compartment rather than the *A* compartment) (dimensionless) | Beta-PERT(0.05;0.15;0.3) | 0.2 (0.08–0.2) | [16,32] |
| $u_p$ | Probability that a newly infected pregnant heifer will develop clinical disease (enter the *I* compartment rather than the *A* compartment) (dimensionless) | Beta-PERT(0.05;0.1;0.3) | 0.1 (0.07–0.2) | [16,32] |
| $M$ | Probability that an individual in the *A* compartment enters the *C* compartment rather than recovering (dimensionless) | N/A | 0.015 | [16] |
| $m$ | Rate at which an individual in the *A* compartment enters the *C* compartment rather than recovering (day$^{-1}$) | $M/D_{All}$ | 0.0007 (0.0003–0.002) | N/A |
| $W$ | Probability that an individual in the *I* compartment enters the *C* compartment rather than recovering (dimensionless) | N/A | 0.18 | [16] |
| $w$ | Rate at which an individual in the *I* compartment enters the *C* compartment rather than recovering (day$^{-1}$) | $W/D_{All}$ | 0.008 (0.004–0.03) | N/A |
| $P_w$ | Probability that a weaned calf from the *I* compartment dies due to *S.* Dublin infection (dimensionless) | Beta-PERT(0.10;0.2;0.5) | 0.2 (0.1–0.4) | [16] |
| $d_w$ | Rate at which a weaned calf from the *I* compartment dies due to *S.* Dublin infection (day$^{-1}$) | $P_w/D_{All}$ | 0.01 (0.04–0.03) | N/A |
| $P_g$ | Probability that a growing heifer from the *I* compartment dies due to *S.* Dublin infection (dimensionless) | Beta-PERT(0.02;0.1;0.3) | 0.1 (0.04–0.2) | [16] |
| $d_g$ | Rate at which a growing heifer from the *I* compartment dies due to *S.* Dublin infection (day$^{-1}$) | $P_g/D_{All}$ | 0.005 (0.002–0.02) | N/A |
| $A$ | Probability of abortion in mid-to-late pregnancy heifers (i.e., raising stage R12) due to *S.* Dublin infection (dimensionless) | Uniform(0.02;0.15) | 0.08 (0.03–0.1) | [16,33] |
| $a_{All}$ | Rate at which *A* and *I* mid-to-late pregnancy heifers (i.e., raising stage R12) abort due to *S.* Dublin infection (day$^{-1}$) | $A/D_{All}$ | 0.004 (0.001–0.01) | N/A |
| $a_C$ | Rate at which *C* mid-to-late pregnancy heifers (i.e., raising stage R12) abort due to *S.* Dublin infection (day$^{-1}$) | $A/D_C$ | 0.0002 (0.00006–0.0004) | N/A |
| $f_n$ | Amount of feces produced by an individual (g day$^{-1}$)[d,e] | N/A | R1: 7,400 | [34,35] |
| | | | R2: 9,200 | |
| | | | R3: 11,000 | |
| | | | R4: 12,700 | |

*(Continued)*

| Notation | Definition (unit) | Equation[a] | Value/Mean (5th, 95th percentile) | Reference |
|---|---|---|---|---|
| | | | R5: 14,500 | |
| | | | R6: 16,300 | |
| | | | R7: 18,100 | |
| | | | R8: 19,900 | |
| | | | R9: 21,700 | |
| | | | R10: 23,500 | |
| | | | R11: 25,200 | |
| | | | R12: 27,000 | |
| $z$ | Number of *S.* Dublin shed in feces by an individual in the *I* compartment (cell g$^{-1}$) | Beta-PERT($10^3$; $10^4$;$10^{5.7}$) | $7.1*10^4$ ($7.8*10^3$–$2.5*10^5$) | [36,37] |
| $\varepsilon_n$ | Number of *S.* Dublin shed in feces by an individual in the *I* compartment (cell day$^{-1}$ individual$^{-1}$)[d] | $f_n$ *$z$ | Estimated from equation for each raising stage | N/A |
| $s$ | Fold-reduction in shedding of *S.* Dublin by individuals in *A* and *C* compared to individuals in *I* (dimensionless) | N/A | 100 | [16] |
| $\lambda_n$ | Number of *S.* Dublin shed by an individual in the *A* or *C* compartment (cell day$^{-1}$ individual$^{-1}$)[d] | $\varepsilon_n$/$s$ | Estimated from equation for each raising stage | N/A |
| $k$ | *S.* Dublin natural decay rate in the $E_n$ compartment (day$^{-1}$) that varies by temperature (T)[d,f] | S1 Appendix | Varies by T, e.g.,: $k=1.25$ at T$=-5$ºC, $k=1.25$ at T$=0$ºC, $k=0.5$ at T$=5$ºC, $k=0.59$ at T$=15$ºC, $k=0.68$ at T$=25$ºC | [38,39] |
| $H$ | Cleaning effectiveness in terms of % of feces removed at each cleaning instance from the $E_n$ compartment (dimensionless)[d] | N/A | 0.95 | Expert elicitation |
| $Cl_{1x/week}$ | % removal of feces from the compartment $E_n$ when cleaning once per week (day$^{-1}$)[d] | S2 Appendix | 0.07 | N/A |
| $\mu$ | Rate of *S.* Dublin removal by cleaning (day$^{-1}$) | Equation 8 | 0.07 | [40] |

[a]Explanation of parameters in distributions: Uniform(min;max); Truncated exponential(lower bound;rate); Beta-PERT(min;mode;max). A Beta-PERT distribution was selected where empirical data were limited but expert knowledge or literature allowed us to define a minimum, most likely, and maximum value.

[b]N/A: Not applicable.

[c]A truncated exponential distribution was used to account for the minimum length of protection against *S.* Dublin.

[d]The subscript $n$ in parameters $f_n$, $\varepsilon_n$, and $\lambda_n$ and compartment $E_n$ were used to refer to the raising stages for a HRO (i.e., weaned heifer: raising stage R1, growing heifer: raising stages R2 to R8, and pregnant heifer: raising stages R9 to R12).

[e]R: Raising stage.

[f]For seasonal variation in average ambient temperature in the Northeastern United States (New York, New Jersey, and Pennsylvania). Temperature range: −5–25°C.

other body areas is considered indirect and mediated by the environment. The direct respiratory transmission of *S.* Dublin is considered possible but uncommon [5], and because its role remains insufficiently studied, it was not included in our model. We therefore consider fecal-oral transmission via the shared environment as the exclusive transmission route in our model [10]. The infected calves progress through three compartments defined based on clinical presentation, *S.* Dublin fecal shedding level, and the length of the infectious period: Asymptomatic (*A*), Clinically ill (*I*), and Carrier (*C*). Upon becoming infected, Susceptible cattle transition into *A* with probabilities 1-$u_w$, 1-$u_g$, or 1-$u_p$ depending on the cattle's age or otherwise move into *I* (with probabilities $u_w$, $u_g$, or $u_p$). The *A* compartment represents individuals experiencing an asymptomatic presentation of *S.* Dublin infection. Weaned calves and growing heifers in the *I* compartment can exhibit severe clinical signs and die at rates $d_w$ and $d_g$ (per day), respectively, due to bacteremia or systemic infection. While in *I* and *A*, infected cattle shed *S.* Dublin in feces at a constant level $\varepsilon_n$ (cell/individual/day) and $\lambda_n$ (cell/individual/day), respectively,

where the subscript $n$ corresponds to the raising stages within a HRO: weaned heifers (R1), growing heifers (R2 to R8), and pregnant heifers (R9 to R12).

Additionally, they either (i) recover from the infection, at a rate $\gamma$ (per day) and enter the Recovered ($R$) compartment where they are temporarily immune to reinfection, or (ii) they develop a carrier status (compartment $C$) at rates $m$ and $w$ (per day) from the $A$ and $I$ compartments, respectively. Individuals in both $A$ and $C$ compartments shed $S$. Dublin at a rate $\lambda_n$, which was considered s-fold (at the baseline s = 100) lower than shedding by an $I$ individual [16]. Eventually, individuals in $C$ recover from infection, at a rate $\eta$ (per day), and enter $R$. Infectious heifers can abort during late pregnancy (i.e., raising stage R12) with rates $a_{All}$ for $A$ and $I$, and $a_C$ for individuals in $C$. An individual becomes once again susceptible to infection after immunity wanes at a rate $\omega$ (per day). The pathogen cells are removed from the environment based on a natural decay rate $k$ (per day) dependent on the seasonally varying average ambient temperature and a barn cleaning rate $\mu$ (per day). Natural mortality among cattle was not included as a parameter as we focused on estimating an unbiased impact of $S$. Dublin on HROs (S1 Fig indicates that including or excluding natural mortality did not influence disease dynamics in the model).

The model accounted for animal aging by dividing each animal compartment into twelve raising stages: i) one for weaned calves (raising stage R1), ii) seven for growing heifers (raising stages R2 to R8), and iii) four for pregnant heifers (raising stages R9 to R12; Fig 1). Accordingly, the environmental compartment $E$ was divided into twelve distinct sections ($E_n$), where the subscript $n$ corresponds to one of the twelve raising stages in an HRO. Each section represents the immediate environment specific to its corresponding raising stage (pen). In this way, the model incorporates spatial structure by dividing animals into distinct pens, allowing for a higher probability of infection spread within pens than between them, with $S$. Dublin spreading across pens at a slower rate ($\kappa$). Individuals progress from one stage to the next every 45 days, with individuals departing from the HRO being replaced by a new batch of fully susceptible calves (batch size corresponding to one-twelfth of the initial herd size $N$). The subscript $n$ (with values 1–12) was also used in model parameters to refer to one of the twelve stages during heifer raising at a HRO.

The initial conditions for the model considered $N$-1 susceptible individuals divided equally across raising stages (i.e., R1: ($N$/12)-1 and R2 to R12: $N$/12 individuals in each stage). After the introduction of an index case in raising stage R1, no additional infectious individuals are imported into the herd. The outcomes of interest predicted in the epidemiological module were (i) across all iterations, the probability of an outbreak (defined as the proportion of model iterations with two or more $S$. Dublin clinical cases) and (ii) at the individual iteration level, $S$. Dublin-induced deaths and abortions during raising, and $S$. Dublin carriers and asymptomatic infections among replacement heifers departing the HRO during the first and second year of the iteration. All outcomes were evaluated under the presence or absence of cleaning improvements and/or vaccination. Model predictions obtained in years beyond the second, representing infection endemic persistence in the HRO, were only minimally different and excluded from this assessment.

## Model equations

The following system of ODEs describes the dynamics of $S$. Dublin transmission for all age categories in a HRO under a baseline scenario where no mitigation strategies have been implemented ($i$ in the equations represents weaned calves, growing heifers, and pregnant heifers, while $j$ refers only to weaned and growing heifers; subscript $n$ refers to one of the twelve progression stages):

$$\frac{dS_i}{dt} = -\beta S_i E_n - \kappa \beta S_i \sum_{n=1}^{12} E_n + \omega R_i \tag{1}$$

$$\frac{dA_i}{dt} = -(\gamma + m)A_i + (1 - u_i)\ \left(\beta S_i E_n + \kappa \beta S_i \sum_{n=1}^{12} E_n\right) \tag{2}$$

$$\frac{dI_i}{dt} = -(\gamma + w + d_j)\, I_i + u_i(\beta S_i E_n + \kappa \beta S_i \sum_{n=1}^{12} E_n)$$

(3)

$$\frac{dC_i}{dt} = -\eta C_i + mA_i + wI_i$$

(4)

$$\frac{dR_i}{dt} = -\omega R_i + \gamma(A_i + I_i) + \eta C_i$$

(5)

$$\frac{dE_n}{dt} = -(k + \mu)E_n + \varepsilon_n I_i + \lambda_n(A_i + C_i)$$

(6)

### Infection control strategies

To evaluate the effectiveness of practices currently used in HROs, such as scraping of the barn's floor and reductions in *S.* Dublin-related mortality with a commercially available vaccine, we implemented in the model different scenarios involving cleaning frequency and vaccination.

   **Vaccination (commercial vaccine).** When vaccination is implemented in the model, all cattle in the herd are assumed to be fully vaccinated and the protective vaccine effectiveness (VE) is observed in all new arrivals from the moment of application of the first vaccine dose (i.e., the start of the simulation). Vaccination provides partial protection, meaning that vaccinated individuals can still experience the consequences of infection (i.e., clinical disease, death, carrier status, and abortion; [41–48]. We defined VE as 1 minus the relative risk in vaccinated animals versus unvaccinated animals (i.e., VE = 1 – risk ratio), with VE values ranging from 0 to 1 (0 = no effectiveness). This approach to assessing vaccination was based on the theoretical study by Lu et al. [20]. Vaccination against *S.* Dublin has been shown to protect vaccinees through a proportional reduction in the probability of death in clinically ill calves ([47]; Table 2, Fig 3). The protection provided by a commercially available vaccine was reported for calves (i.e., 2–5 weeks old), but due to the absence of information, here we extrapolated its effect to older cattle (i.e., weaned calves and growing heifers in the model), and then evaluated the impact of this assumption in sensitivity analysis. Some other effects, such as limiting the development of clinical symptoms, the carrier state in infected cattle, and fecal shedding, have been observed with experimental vaccines, but no similar effects have been reported for vaccines commercially available for use in cattle ([41–56], Fig 3). It is important to note that, despite the variety of vaccine effects reported in the literature, genomic and antigenic heterogeneity among *Salmonella* serovars may influence vaccine effectiveness, as most vaccines are developed using specific laboratory strains that may not represent the full diversity present in the field [57,58]. In the vaccination scenario, both weaned and growing heifers are vaccinated with the commercial vaccine at the beginning of the simulation, with vaccines also being administered to new batches of weaned calves upon arrival to a HRO. Growing heifers are revaccinated a year after their initial immunization as weaned heifers. Protection in vaccinated calves persists for their entire time in the HRO. The vaccine effectiveness against *S.* Dublin was represented in the model as a value reducing the probability of death after clinical disease ($d_j$) as follows:

$$d_j * (1 - VE)$$

(7)

Where the subscript "*j*" refers to either weaned calves (*w*) or growing (*g*) heifers.

**Table 2. Parameter notations, definitions, and values related to cleaning improvements and vaccination with a commercial vaccine as control strategies against *Salmonella* Dublin spread in a heifer-raising operation.**

| Notation | Definition (unit) | Equation | Mean (5th, 95th percentile) | Reference |
|---|---|---|---|---|
| ***Vaccination (commercial vaccine)*** | | | | |
| $VE$ | Vaccine effectiveness in reducing the probability of death after clinical disease (dimensionless) | $VE = 1-$ Beta-PERT$(0.05;0.2;0.8)$[a] | 0.72 (0.48–0.91) | [47] |
| ***Improved cleaning*** | | | | |
| $CI_{3x/week}$ | Average % of feces removed daily from the Environment compartment after cleaning three times per week (day$^{-1}$)[b] | S2 Appendix | 0.20 | N/A[b] |
| $CI_{5x/week}$ | Average % of feces removed daily from the Environment compartment after cleaning five times per week (day$^{-1}$) | S2 Appendix | 0.33 | N/A |
| $CI_{7x/week}$ | Average % of feces removed daily from the Environment compartment after cleaning seven times per week (day$^{-1}$) | S2 Appendix | 0.47 | N/A |
| $CI_{2x/day}$ | Average % of feces removed daily from the Environment compartment after cleaning two times per day (day$^{-1}$) | S2 Appendix | 0.74 | N/A |
| $CI_{4x/day}$ | Average % of feces removed daily from the Environment compartment after cleaning four times per day (day$^{-1}$) | S2 Appendix | 0.87 | N/A |
| $CI_{6x/day}$ | Average % of feces removed daily from the Environment compartment after cleaning six times per day (day$^{-1}$) | S2 Appendix | 0.91 | N/A |
| $CI_{8x/day}$ | Average % of feces removed daily from the Environment compartment after cleaning eight times per day (day$^{-1}$) | S2 Appendix | 0.93 | N/A |
| $CI_{12x/day}$ | Average % of feces removed daily from the Environment compartment after cleaning twelve times per day (day$^{-1}$) | S2 Appendix | 0.96 | N/A |

[a]Explanation of parameters in distribution: Beta-PERT(min;mode;max). Parameter $VE$ can only vary from 0 to 1. In the baseline model, $VE = 0$.

[b]N/A: Not applicable.

| Reference | Age category | Vaccine type | Probability of infection after exposure | Probability of developing symptoms | Probability of death if infected | Probability of developing a carrier state | Duration of the infectious period | Fecal shedding | In utero transmission to calf |
|---|---|---|---|---|---|---|---|---|---|
| **Experimental vaccines** | | | | | | | | | |
| Smith et al. 1980 [49] | Calves | Modified-live | — | ✗ | ✗ | — | — | — | — |
| Robertsson et al. 1983 [50] | Calves | Modified-live | — | ✓ | ✓ | ✓ | — | ✓ | — |
| | | Inactivated | — | ✗ | ✓ | — | — | ✓ | — |
| Smith et al. 1984 [51] | Calves | Modified-live | — | — | ✓ | — | — | — | — |
| Smith et al. 1984 [52] | Calves | Modified-live | — | ✗ | ✓ | — | — | — | — |
| Jones et al. 1991 [53] | Calves | Modified-live | — | — | ✓ | — | — | — | — |
| Mukkur et al. 1991 [54] | Calves | Modified-live | — | — | ✓ | — | — | — | — |
| Smith et al. 1993 [55] | Calves | Modified-live | — | ✓ | ✓ | — | — | — | — |
| Selim et al. 1995 [56] | Calves | Modified-live | — | ✗ | ✓ | — | — | — | — |
| **Commercial vaccines** | | | | | | | | | |
| Fox et al. 1997 *[41] | Calves | Modified-live | — | — | — | ✓ | ✗ | ✓ | — |
| House et al. 2001 *[42] | Cows | Modified-live | ✗ | — | ✗ | — | — | ✓ | — |
| Heider et al. 2008 [43] | Cows | SRP | ✗ | — | — | — | ✗ | — | — |
| Hermesch et al. 2008 [44] | Cows | SRP | ✗ | ✗ | — | — | ✓ | — | — |
| Dodd et al. 2011 [45] | Cows | SRP | ✗ | — | ✗ | — | — | — | — |
| Habing et al. 2011 [46] | Calves and heifers | Modified-live | — | ✗ | ✗ | — | — | — | — |
| Cummings et al. 2019 [47] | Calves | Modified-live | ✗ | ✗ | ✓ | — | — | — | — |
| Castro-Vargas et al. 2024 [48] | Cows | Modified-live | — | — | — | — | — | — | ✓ |

**Legend:** ✓ Vaccine was protective ✗ Vaccine was not protective — Vaccine effect was not evaluated

SRP: Siderophore receptor and porin protein vaccine

*The vaccine is not commercially available for use in cattle.

Reference numbers are indicated in square brackets [ ].

**Fig 3. Experimental and commercial vaccine effects against *Salmonella* spp. and *S*. Dublin infection in cattle assessed and reported in the literature.**

**Improved cleaning.** The cleaning effectiveness per cleaning instance ($H$), understood as the proportion of feces that is removed from the barn every time the barn is scraped was determined based on expert elicitation from two veterinarians with more than 10 years of experience in dairy production medicine and bovine veterinary practice, one academic in dairy sciences, and one dairy management professional. Experts were asked about the perceived percentage of feces removed from the barn's floor after one cleaning instance (i.e., the value of $H$) on a HRO or dairy farm they are familiar with, to which, on average, they responded that about 95% of feces (range: 90% to 99%) are removed. It is important to note that $H$ does not describe the effectiveness of barn cleaning each day, as cattle will continue to produce feces immediately after every cleaning instance. The overall proportion of feces removed each day from the barn floor (referred to as the daily cleaning effectiveness, $Cl_{freq}$) was approximated by averaging the maximum (present just before cleaning) and minimum (present just after cleaning) amount of feces on the barn floor and bedding area (i.e., cow mattress) during the day for a particular cleaning frequency (calculation modified from Gautam et al. [40]). The calculation considered the amount of feces produced by an individual, the number of individuals in a given age category in the herd, and the cleaning frequency per day (S2 Appendix). Then, the cleaning effectiveness $Cl_{1/week}$ represents the proportion of feces removed from compartment E when cleaning once per week (day$^{-1}$).

Cleaning practices were assessed in eight scenarios (Table 2). In three of these scenarios, feces on the barn floor were cleaned multiple times per week: three times (3x) per week ($Cl_{3x/week}$), 5x per week ($Cl_{5x/week}$), and 7x per week ($Cl_{7x/week}$). The remaining five scenarios considered cleaning multiple times per day: 2x per day ($Cl_{2x/day}$), 4x per day ($Cl_{4x/day}$), 6x per day ($Cl_{6x/day}$), 8x per day ($Cl_{8x/day}$), and 12x per day ($Cl_{12x/day}$). The per week cleaning frequencies are achievable using a skid steer, while more frequent cleaning, as assessed in the study, would represent the use of an alley scraper. An additional scenario was included to assess *S.* Dublin dynamics in a bedded pack barn in which bedding is applied 1x per day and then completely replaced every 4 weeks. We assumed that bedding addition covers 95% of the manure present in the pen (i.e., equivalent to 95% cleaning effectiveness). The daily cleaning effectiveness ($Cl_{freq}$), was converted into the rate $\mu$ (day$^{-1}$) at which *S.* Dublin is removed every day from the barn environment using the following formula described in Gautam et al. [40]:

$$\mu = -\ln\left(1 - Cl_{freq}\right) \tag{8}$$

Where subscript "*freq*" refers to cleaning frequencies between 1x/week to 12x/day. When cleaning was performed on a weekly basis, the rate for cleaning once per day was adjusted by multiplying by 1/7, 3/7, or 5/7 for cleaning 1x, 3x, or 5x per week, respectively. Equation 8 is based on the exponential distribution formula used for converting the probability of an event into the rate of the event (which is needed to model the event in an ODE model). While probability is bounded by 1, the corresponding rate is unbounded; thus, as $Cl_{freq}$ approaches 1, $\mu$ increases rapidly, approaching infinity. It should be noted that $Cl_{freq}$ is a function of both the cleaning effectiveness per cleaning instance ($H$) and cleaning frequency. Similar $Cl_{freq}$ can be achieved with different combinations of the two parameters (e.g., a low H can be compensated by higher cleaning frequency and the opposite). Importantly, increases in cleaning frequency lead to larger gains in $Cl_{freq}$ at lower frequencies, while improvements become smaller as the frequency increases reflecting diminishing returns.

### Economic module

**HRO's operating costs, income, and operating income.** The economic module was based on partial budgeting. It takes the predictions from the epidemiological module to estimate the farm operating income, calculated as the difference between the income from raising heifers and the operating costs for running a HRO. The income encompassed the earnings from pregnant heifers returning to their farms of origin (*TIn*) and from the sale of apparently healthy open heifers for slaughter (i.e., heifers that aborted during mid-to-late pregnancy and prior to their departure from the HRO; *AIn*; Fig 4, Table 3). The number of open heifers sold was estimated as the cumulative number of *S.* Dublin-induced abortions

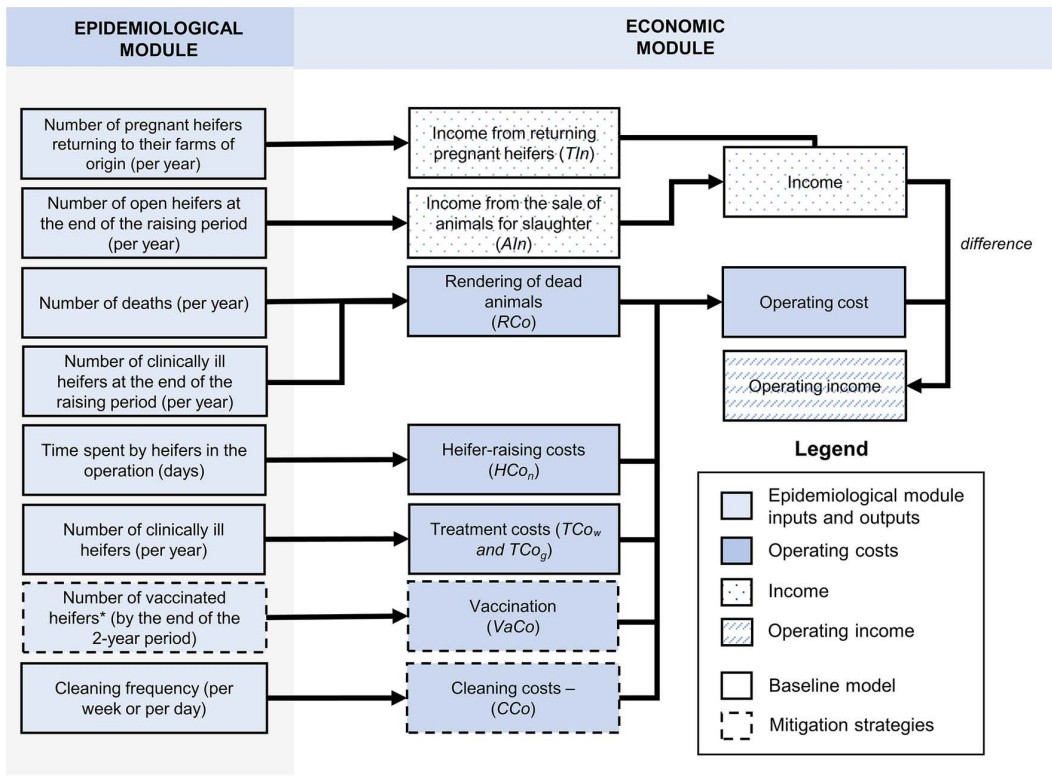

**Fig 4. Components of the epidemiological and economic modules determine the operating cost, income, and operating income of a HRO.** Inputs and outputs from the epidemiological module are used to calculate the operating cost, income, and operating income under a baseline "do-nothing" scenario (baseline level of cleaning (1x per week) and no vaccination) and scenarios with the implementation of cleaning improvements and/or vaccination.

by the end of the 2-year simulation period. The 2-year simulation length was selected because *S.* Dublin dynamics show no remarkable variations after two years into the simulation, while selecting a shorter period provides a narrower understanding of *S.* Dublin epidemiological and economic impacts on a HRO.

The operating costs were based on the daily cost of heifer-raising (including labor, veterinary treatment unrelated to *S.* Dublin, feed, equipment maintenance, and bedding; $HCo_n$), additional treatment costs in clinically ill individuals ($TCo_w$ and $TCo_g$), and rendering of dead animals ($RCo$; Fig 4, Table 3). Treatment costs were included based on prices from The Cornell Ambulatory and Production Service Clinic provided by F.A.L.Y, a veterinarian and epidemiologist with more than 10 years of experience working with dairy cattle and raising cattle operations. $HCo_n$ varied according to the raising stage (subindex *n* in $HCo_n$ represents raising stages from R1 to R12) and decreased with increasing herd size through a logarithmic function (Table 3) [59,60]. Rendering dead animals results in costs ($RCo$) without generating any income for the HRO. If vaccination was carried out, the vaccination cost ($VaCo$) covering labor and doses was included. For cleaning, costs were considered for each additional barn floor cleaning per week or per day ($CCo$).

The income derived from raising a heifer ($DIn$) was obtained based on expert opinion from the same veterinarian and epidemiologist who provided information on treatment costs (F.A.L.Y). The operating income was expressed as USD/100-head by the end of the first and second years into the simulation. The proportional change in operating income was calculated by comparing operating incomes for a specific control strategy and a "do-nothing" approach (baseline level of cleaning (1x per week) and no vaccination) over the 2-year simulation period. The consequences of milk yield reductions due to *S.* Dublin infection in the operating income were not assessed as heifers calve after leaving the HRO.

**Table 3. Details about the economic module parameters used to calculate operating costs and income when no mitigation strategies have been implemented (baseline). Operating costs and income are in USD. The number of cattle on the farm (*N*) is 1,000 for the baseline scenario.**

| Notation | Definition | Formula | Value | Reference |
|---|---|---|---|---|
| ***Operating costs*** | | | | |
| *Heifer raising-related* | | | | |
| $HCo_n$[a] | Heifer-raising costs (USD/day) | R1: (−0.27*ln(N)+ 3.97)*0.85 | R1: 1.8 | [59,60] |
| | | R2: (−0.27*ln(N)+ 3.97)*0.79 | R2: 1.7 | |
| | | R3: (−0.27*ln(N)+ 3.97)*0.72 | R3: 1.5 | |
| | | R4: (−0.27*ln(N)+ 3.97)*0.72 | R4: 1.5 | |
| | | R5: (−0.27*ln(N)+ 3.97)*0.76 | R5: 1.6 | |
| | | R6: (−0.27*ln(N)+ 3.97)*0.81 | R6: 1.7 | |
| | | R7: (−0.27*ln(N)+ 3.97)*0.88 | R7: 1.9 | |
| | | R8: (−0.27*ln(N)+ 3.97)*0.92 | R8: 1.9 | |
| | | R9: (−0.27*ln(N)+ 3.97)*0.90 | R9: 1.9 | |
| | | R10: (−0.27*ln(N)+ 3.97)*0.88 | R10: 1.9 | |
| | | R11: (−0.27*ln(N)+ 3.97)*0.89 | R11: 1.9 | |
| | | R12: (−0.27*ln(N)+ 3.97)*1.0 | R12: 2.1 | |
| *Salmonella Dublin-related* | | | | |
| $TCo_w$ | Treatment of clinically ill weaned calves (USD/individual) | N/A[b] | 79.7 | Expert elicitation |
| $TCo_g$ | Treatment of clinically ill growing and pregnant heifer (USD/individual) | N/A | 99.8 | Expert elicitation |
| *RCo* | Rendering of dead animals (USD/individual) | N/A | 125 | Commercial rendering services |
| ***Income*** | | | | |
| *DIn* | Income for raising a heifer (USD/heifer/day) | N/A | 2.5 | Expert elicitation |
| *TIn* | Total income received from a heifer that returned pregnant to its facility of origin (USD/heifer) | DIn*540 | 1,350 | Estimated |
| *Car* | Carcass value (USD/cwt) | N/A | 125 | [61] |
| *Dress* | Dressing percentage of cattle (dimensionless) | N/A | 0.63 | [62] |
| *Weight* | Weight of animals at slaughter (cwt)[c] | N/A | 9.7 | [63] |
| *AIn* | Income from the sale of animals for slaughter (USD/cwt) | Car*Dress*Weight | 1,070 | [61] |

[a]The subscript *n* refers to the raising stages (R) for a HRO (i.e., calf: raising stage R1, growing heifer: raising stages R2 to R8, and pregnant heifer: raising stages R9 to R12).

[b]N/A: Not applicable.

[c]Weight estimation based on a 630-day-old heifer.

The module considers that calves enter the operation at 90 days of age (weaning age) and become pregnant after 360 days in the operation (at ~15 months of age; [64]). In cases where cattle died prior to returning to their farms of origin, the operating cost was calculated based on the days spent in the operation until the raising stage prior to their death, plus half of the duration of the raising stage at the time of their death. For example, if an individual died in the seventh raising stage, it would mean that it completed raising stages from one to six (270 days) plus half of the seventh raising stage (22.5 days), adding up to a total of 292.5 days in the HRO.

## Operating income under different cost scenarios

Given the lack of information regarding the specific costs of applying vaccination and increasing cleaning frequency, a scenario analysis was carried out to determine the operating income of a HRO under different hypothetical costs of a mitigation strategy. We only evaluated the economic impact of a commercial vaccine that reduces *S*. Dublin-related mortality

[47] in calves and heifers; no other vaccination effects were included in the economic analysis. For the cost assessment of vaccination, we considered that two doses are required for full vaccination status during the first year (i.e., two doses to effectively vaccinate each individual) and revaccination with a single dose after one year in the HRO. The cost for each of the cleaning scenarios represents the hypothetical cost (energy, labor, equipment maintenance) of an additional cleaning event beyond the cost at the frequency of cleaning once per week (baseline frequency). Estimations from the economic assessment were corroborated by the veterinarian expert in dairy farming who provided information on treatment costs and income (F.A.L.Y).

### Model running

We introduced stochasticity into the model through Monte Carlo simulation. This approach involves running multiple iterations of the model repeatedly and, in each iteration, specific values for parameters are randomly drawn from specified probability distributions to mimic the variation found in natural biological systems [65]. Each simulation consisted of 1,000 iterations. This number of iterations was chosen based on recommendations in Winston [66] and because the mean and coefficient of variation (CV) of outcomes (the cumulative numbers of individuals in $A$ and $I$ compartments, carriers, and recovered individuals by the end of a 2-year long simulation) were similar under 5,000 and 1,000 iterations (i.e., the differences in the mean number of individuals were within ±2 and the CV was within ±0.5; S1 Table). The R script and accompanying notes are available for open use at: https://github.com/IvanekLab/SDublin.git

### Model validation

Since there are no available reports on *S*. Dublin spread in HROs, we validated the model by comparing predicted seroprevalence with data from a dairy farm in Tennessee affected by *S*. Dublin [67]. Predictions were obtained for multiple scenarios in which a different assumed number of infected pregnant cattle entered the operation to reflect the different possibilities of infection introduction into this Tennessee dairy farm. Predictions from the model were considered valid if its interquartile range (IQR) encompassed the true seroprevalence value and its median predicted seroprevalence value was within the 95% confidence interval around the average true seroprevalence (S1 Appendix).

### Sensitivity analysis

Sensitivity analysis was done using Partial Rank Correlation Coefficient (PRCC) with the R package *epiR* [68]. The analysis examined the relationship between input parameters ($D_{All}$, $D_c$, $D_R$, $d_w$, $d_g$, $a_{All}$, $a_C$, $u_c$, $u_g$, $u_p$, and $z$) and outcomes from the epidemiological and economic modules by the end of a 2-year simulation period. PRCC allows for determining the strength and direction of a certain parameter's association over a model outcome of interest while controlling for the effect of other parameters included in the model [69]. Correlations were considered statistically significant based on Bonferroni-corrected *p*-value (significance at $p < 0.005$ estimated as 0.05/11 model parameters) to reduce type-I errors (e.g., [70]) and only correlation coefficient ($\rho$) values $> |\rho| = 0.4$ were deemed relevant for reporting (e.g., [71]). Results from the PRCC analysis were visualized using a heat map. Classification trees were built in R using the *caret* package [72] to understand better the influence of the interaction among model parameters on the probability of an outbreak (defined as the presence or absence of >= 2 clinical cases over the length of the simulation). A 10-fold cross-validation procedure was carried out to assess the classification model performance (i.e., accuracy and agreement between predicted and expected classifications). Tree post-pruning was performed using the Cost-Complexity Pruning approach to achieve the simplest yet best-performing tree that accurately represents the simulated data [73].

### Scenario analysis

We evaluated scenarios involving the implementation of a commercial vaccine reducing *S*. Dublin-related mortality and improvements in cleaning practices on the epidemiological and economic impacts of *S*. Dublin. These include eight

cleaning frequency scenarios (cleaning 3x/week to cleaning 12x/day), one vaccination scenario (commercial vaccine), and eight scenarios assessing the combination of cleaning frequencies and vaccination. In addition to the reduction in the probability of death through vaccination, we also considered hypothetical vaccine effects that have not yet been demonstrated for existing commercial *S.* Dublin vaccines but could be valuable to assess to inform future vaccine improvements. These include 25%, 50%, and 75% VE in reducing (i) susceptibility to *S.* Dublin, (ii) length of the infectious period, and (iii) shedding, as follows: $\beta * (1 - VE)$ for reductions in susceptibility, $D_{A/I} * (1 - VE)$ for reductions in the infectious period, and $\varepsilon_n * (1 - VE)$ for shedding reduction in clinically ill and $\lambda_n * (1 - VE)$ for shedding reductions in asymptomatic and carriers; where the subscript "n" represents a specific raising stage. For these hypothetical vaccination scenarios, we only estimated their effectiveness in deaths, abortions, and asymptomatic and carrier departures, but we did not include them in the economic analysis.

We also assessed seasonal variation (e.g., temperature effects), herd sizes, and reductions in shedding from asymptomatic and carrier animals compared to clinically ill. Additional scenarios explored the presence or absence of shedding across infectious states, the effectiveness of cleaning at each instance, and the introduction of different numbers of asymptomatic and/or carrier calves at the start of the simulation (t = 0). The importance of uncertainty in the transmission rate, the probability of carrier development in the asymptomatic compartment, and the probability of carrier development among clinically ill animals was addressed by assessing different values for these parameters (S2 Table).

## Results

The developed model successfully replicated the 41% true seroprevalence estimated from the apparent prevalence in Kent et al. [67], assuming testing at 20, 25, 30, 35, or 40 days post the initial introduction of a single or multiple asymptomatic and carrier pregnant heifers (S1 Appendix), thus supporting the validity of the model.

### *Salmonella* Dublin shedding amount in feces, infectious period, and time spent in the Recovered compartment influenced epidemiological outcomes and the operating income the most, whereas the outbreak probability was mostly determined by the interaction between the *S.* Dublin shedding rate and duration of the infectious period

Among the most influential parameters assessed in the sensitivity analysis, *S.* Dublin shedding amount in feces (*z*) and the duration of the infectious period ($D_{All}$) were positively correlated with all epidemiological outcomes (deaths, abortions, carrier departures, and asymptomatic departures) and negatively correlated with the operating income (Fig 5). A strong correlation in the opposite direction was observed for the duration in the *R* compartment ($D_R$) as it was negatively correlated with epidemiological outcomes and positively correlated with the operating income (Fig 5). Findings from the classification tree highlight the interaction between the *S.* Dublin shedding amount in feces and the duration of the infectious period as the primary predictors for the probability of an *S.* Dublin outbreak following the introduction of an asymptomatic heifer into a HRO (Fig 6). Specifically, the classification tree indicated that there was a < 1% probability of an outbreak if the *S.* Dublin shedding amount was < $2.2*10^3$ cells/g of feces and the duration of the infectious period was less than 32 days or if the duration of the infectious period was < 9.1 days and the shedding amount was < $5.5*10^3$ cells/g.

### Vaccination with a commercial vaccine against *Salmonella* Dublin reduces deaths while increasing the cleaning frequency reduces all values for epidemiological outcomes

Findings indicate a 79% (785/1,000 iterations) probability of an outbreak following the introduction of an asymptomatic individual into a HRO housing 1,000 heifers when cleaning is performed once per week (baseline model; S3 Table). Within the first year of the simulation, HROs affected by *S.* Dublin experienced medians of 20 deaths (IQR = 9–35), 10 abortions (IQR = 5–15), 15 carrier replacement heifers (IQR = 11–19), and 57 asymptomatic replacement heifers (IQR = 34–84) (Fig 7, S3 Table). Vaccination with a commercial vaccine reducing *S.* Dublin-related mortality was a very effective

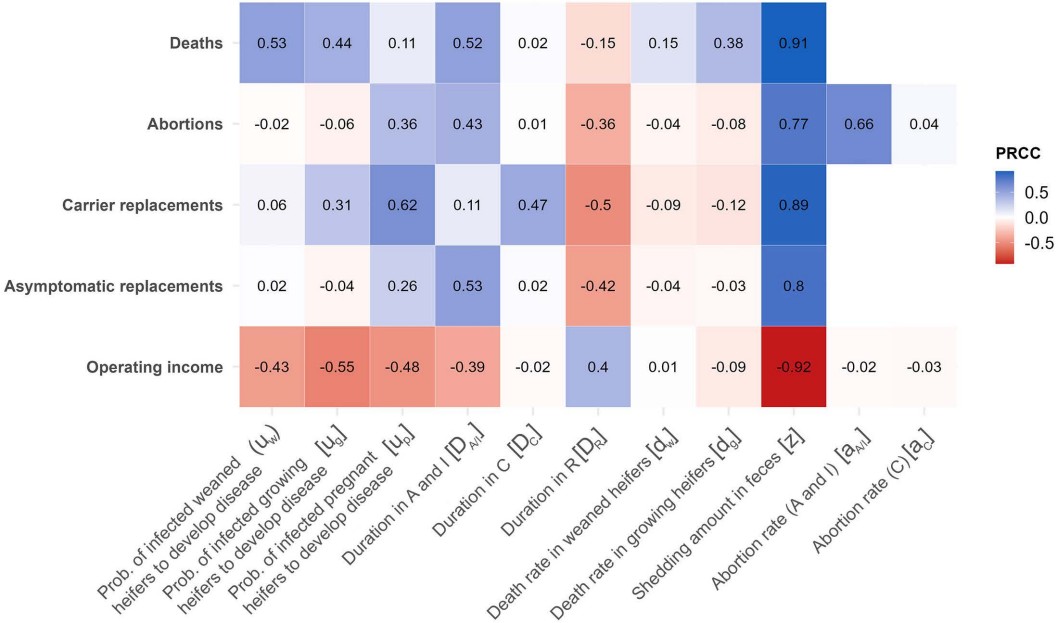

**Fig 5. Findings from the sensitivity analysis.** Heat map showing the Partial Rank Correlation Coefficient (PRCC) for the correlation between the model parameters (columns) and outcomes (rows), namely, deaths, abortions, and carrier and asymptomatic replacement heifers, and the operating income of a heifer-raising operation (Bonferroni corrected-$p<0.005$; $|\rho|>0.4$). $A$ = asymptomatic, $I$ = clinically ill, $C$ = carrier, and $R$ = recovered.

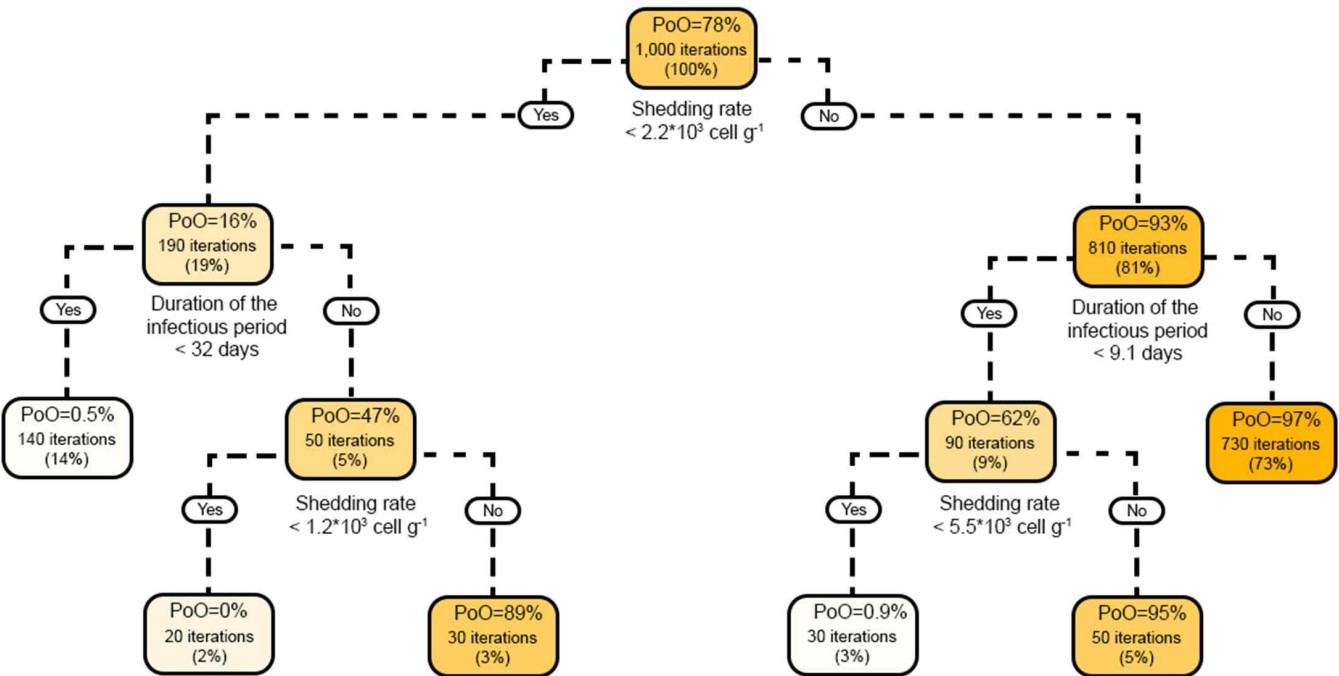

**Fig 6. Classification tree of the probability of an outbreak after the introduction of an asymptomatic index case at time $t=0$.** PoO = probability of an outbreak (the proportion of iterations with two or more *S. Dublin* clinical cases following the introduction of an asymptomatic index case). Higher color saturation in the boxes indicates an increased probability of an outbreak.

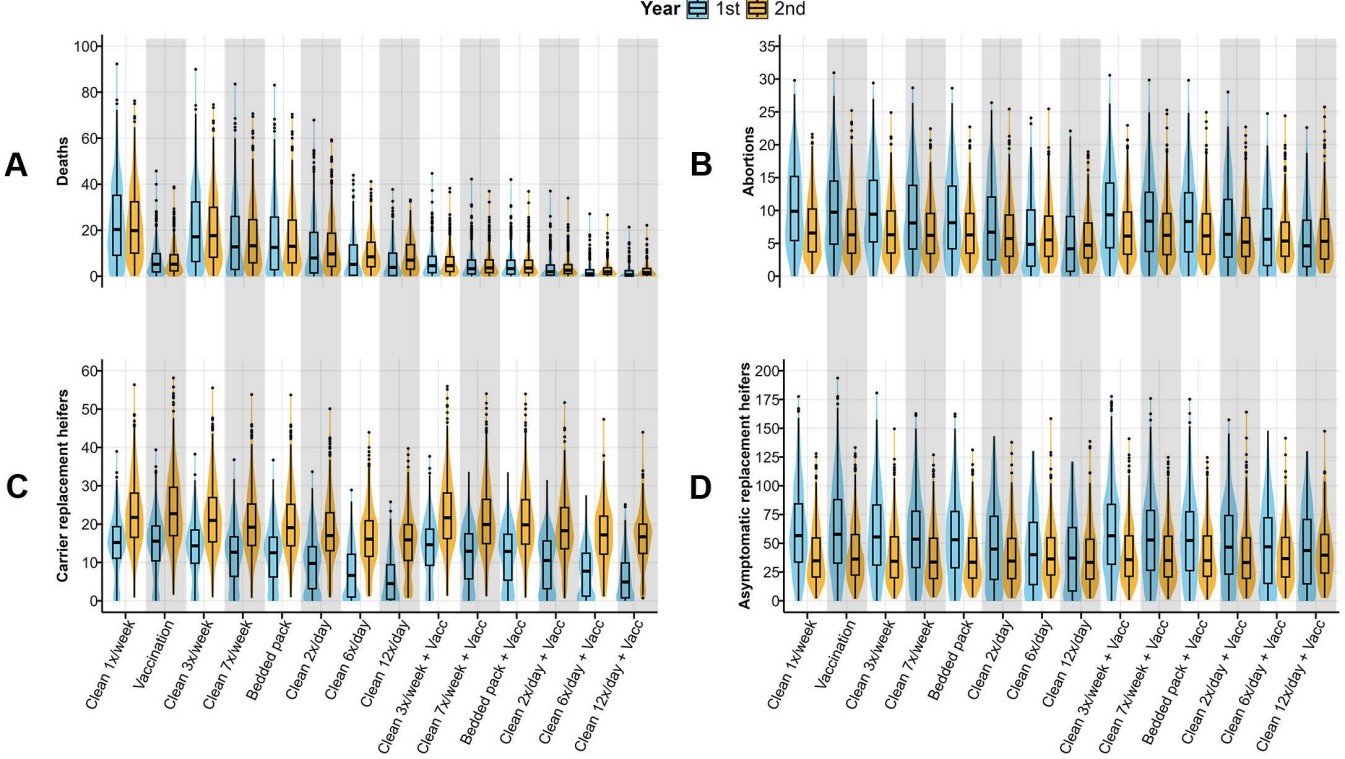

**Fig 7.** *Salmonella* Dublin-related (A) deaths and (B) abortions and (C) carrier and (D) asymptomatic infections in replacement heifers observed in scenarios with and without the implementation of vaccination (with a commercial vaccine) or/and increased cleaning frequency in an HRO with N = 1,000 heifers at the simulation start. A comparison was made between the baseline scenario (baseline level of cleaning (1x per week) and no vaccination) and scenarios in which control strategies were implemented. These strategies included cleaning at different frequencies per week (3x, 7x) and per day (2x, 6x, and 12x) and/or vaccination (with a commercial vaccine) reducing mortality due to *S.* Dublin (vaccine effectiveness: median = 0.72, 5th–95th percentile = 0.48–0.91). Horizontal blue and orange lines in each plot indicate the median value of the corresponding outcome under the baseline scenario.

standalone control measure in decreasing median deaths, with a 75% reduction (median = 5 deaths, IQR = 2–10) compared to the baseline scenario. Cleaning at the highest frequency assessed (12x per day) reduced the outbreak probability to 23% (234/1,000 iterations), median deaths to 4 (80% reduction, IQR = 0–10), median abortions to 4 (60% reduction, IQR = 1–9), median carrier replacement heifers to 5 (67% reduction, IQR = 0–9), and median asymptomatic replacement heifers to 37 (35% reduction, IQR = 9–64). The scenario combining improvements in cleaning (12x per day) and vaccination showed a meaningful reduction in deaths (95% reduction, median = 1, IQR = 0–2), abortions (50% reduction, median = 5, IQR = 2–9), and carrier replacement heifers (67% reduction, median = 5, IQR = 1–10), with a less pronounced reduction in asymptomatic replacement heifers (23% reduction, median = 44, IQR = 15–71) compared to the baseline. It is important to note that scenarios involving mortality reduction via vaccination showed a slight increase in carrier and asymptomatic infections, thereby marginally countering the impact of cleaning improvements on preventing infection spread to other operations ([Fig 7](), [S3 Table]()).

Findings from the scenario analysis of hypothetical vaccines showed that vaccine-induced susceptibility reduction and shedding reduction produced identical epidemiological outcomes, as both acted on complementary components of transmission dynamics in our model equations ([Equations 1](), [2](), [3]() and [6]()). For these scenarios, small vaccine-driven reductions of 25% in susceptibility and shedding led to approximately 20% fewer deaths (median = 19; IQR = 8–35) compared to a "no vaccination" scenario by the end of the first year. Furthermore, carrier replacement heifers decreased by about 7%

(median = 14; IQR = 8–18) with a 25% reduction in susceptibility and by up to 67% (median = 5; IQR = 1–12) with a 75% reduction by the end of the first year due to the lower number of infectious animals present at the same time in the HRO. Shortening the infectious period produced the largest impact on asymptomatic replacements, with reductions of up to 72% (median = 16; IQR = 9–23). The median number of abortions declined by about 10% (median = 9; IQR = 5–14) at a 25% reduction in susceptibility and shedding (S3 Table). Impacts of these hypothetical vaccine effects were similar between the first and second year.

**Stringent cleaning can decrease the impact of *S.* Dublin on the operating income of a HRO, even when it carries a high cost of implementation**

In the absence of *S.* Dublin infection, a HRO with 1,000 heifers achieves an operating income of USD 39,123 per 100-head raised each year of production. This value decreases to a median of USD 32,827 (IQR = 30,650–34,875) and USD 33,517 (IQR = 31,443–35,179) per 100-head for years one and two, respectively (i.e., a median reduction from the initial operating income of 16% for year one and 14% for year two) after an infectious calf introduces *S.* Dublin into the herd and no mitigation strategies are implemented. Findings from our economic module indicate that for the cost scenarios considered, increasing cleaning frequency from 1x/week to 7x/week can increase the median operating income of a HRO post *S.* Dublin introduction between 2.7% to 3.8% during the first year (Fig 8, S4 Table). Meanwhile, increasing the cleaning frequency to 12x/day (the most stringent cleaning evaluated in our study) can increase the median operating income of a HRO by 1.2% to 10.6% if the cost of additional daily cleaning per 100-head does not surpass USD 0.45 (Fig 8, S4 Table). Furthermore, vaccination with an available commercial vaccine can increase the operating income by 1.2% to 2.4% compared to a "do-nothing" scenario during the first year only if costs per 100 vaccine doses do not surpass USD 100 (Fig 8, S4 Table). Stringent cleaning measures combined with vaccination with an available commercial vaccine, did not improve the operating income in most cost scenarios (maximum operating income increase of 11.3% during the first year when vaccination does not require an additional investment; S4 Table, S2 Fig).

Findings from the scenario analysis also indicate that introducing multiple infectious index cases into the HRO at time t = 0 did not notably increase the risk and consequences of *S.* Dublin infection compared to a single index case introduction. After the introduction of a single or multiple asymptomatic or carrier individuals at time t = 0, the likelihood of a *S.* Dublin outbreak occurring when no mitigation strategies were in place was almost identical to the baseline scenario (S3 Table). Likewise, only very minor differences were observed in other epidemiological outcomes (S3 Table). In contrast, reduced shedding levels in asymptomatic and carrier individuals relative to clinically ill meaningfully influenced all epidemiological outcomes (S3 Table). Furthermore, clinical cases are critical to triggering a *S.* Dublin outbreak and its epidemiological consequences in HROs (S3 Table).

## Discussion

We developed a mathematical model that integrates available knowledge of the epidemiology of *S.* Dublin in HROs and conducted an *in silico* assessment of the effectiveness of cleaning improvements and vaccination with an available commercial vaccine as control strategies. *S.* Dublin isolates from cattle in US dairy farms have displayed resistance to multiple antibiotics [7,47,74]. Considering that no differences in transmission between susceptible and MDR *S.* Dublin have been identified to date, the model presented in our study can be considered to represent the transmission of both susceptible and MDR *S.* Dublin strains. However, economic impacts would likely be larger for MDR strains given the higher treatment costs and potentially prolonged infections associated with MDR strains. The main study findings are: (i) cleaning multiple times per day, with cleaning 12x/day being the most effective, meaningfully reduced the probability of *S.* Dublin introduction into HROs and its consequences, while also improving the operating income of these operations depending on the implementation costs; (ii) vaccination with an available commercial vaccine greatly prevents deaths from *S.* Dublin

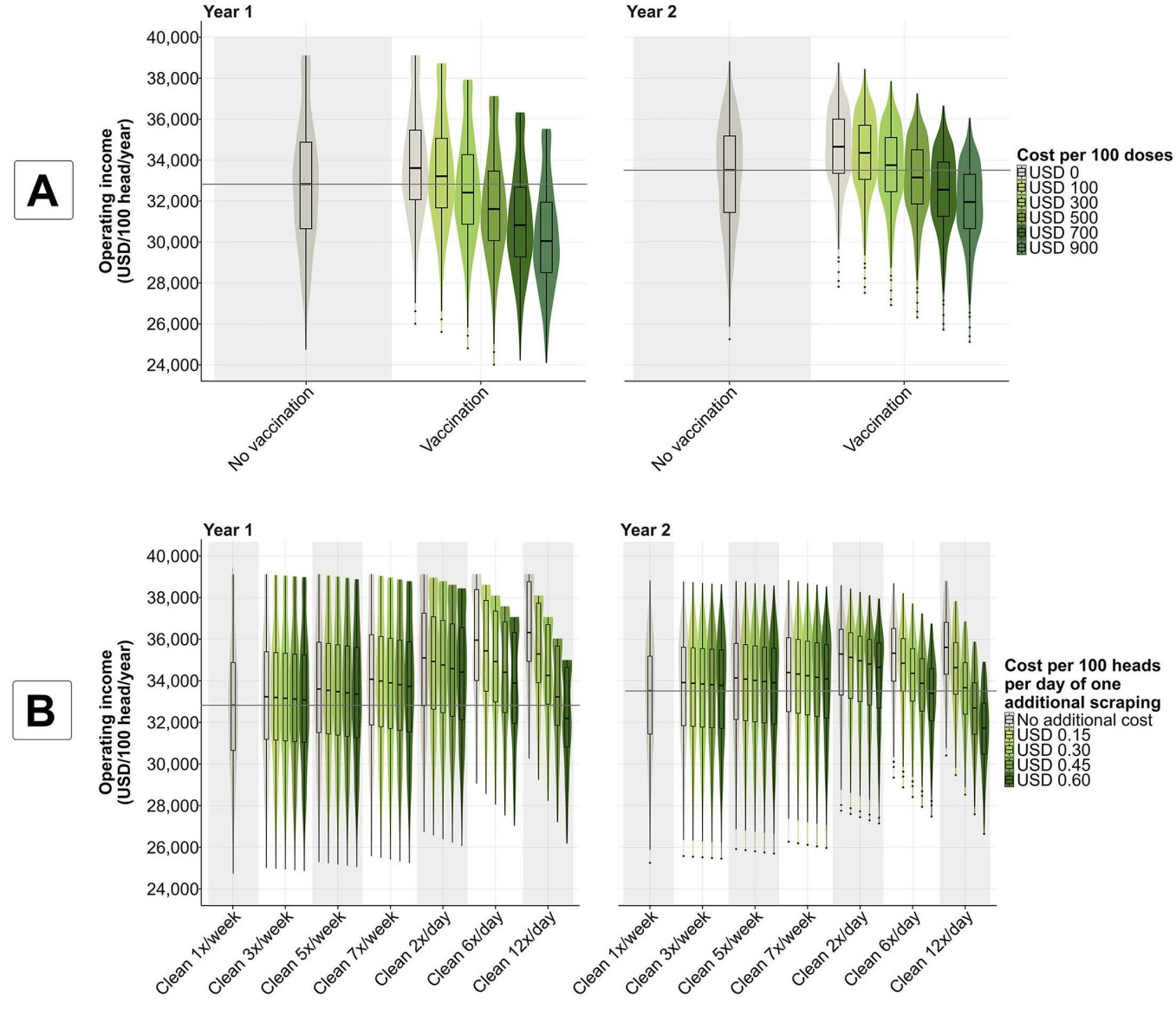

**Fig 8. Operating income (USD per 100-head) observed in scenarios with different costs required for the implementation of mitigation strategies.** A comparison was made between the baseline scenario (cleaning once (1x) per week and no vaccination), and thus no additional costs, and scenarios in which control strategies were implemented and compared in terms of the operating income (USD per 100-head). These strategies included (A) vaccination reducing *S*. Dublin-related mortality (vaccine effectiveness: median = 0.72, 5th-95th percentile = 0.48–0.91) and (B) cleaning at different frequencies (3x, 5x, and 7x times per week and 2x, 6x, and 12x per day). Findings were divided into the first year and the second year of the simulation. Operating income in a "no infection" scenario = USD 39,123 per 100 head each year of the simulation. Horizontal gray line in each plot-year indicates the median value of the operating income under the baseline (cleaning once (1x) per week and no vaccination).

infection, but does not prevent its transmission; and (iii) *S*. Dublin presence in HROs can be sustained through the continuous arrival of naïve individuals, not requiring the repeated introduction of carriers, while the occurrence of clinical disease in infected heifers promotes *S*. Dublin spread in the operation. Our findings also imply that HROs play a meaningful role in the transmission of *S*. Dublin to dairy cattle herds. The implications of these results, along with the study's limitations, are discussed in the following sections.

## Novel mathematical model of *S*. Dublin transmission and its implications within the context of HROs

Our mathematical model of the transmission dynamics of *S*. Dublin in a US HRO represents a novel addition to an array of models developed previously for *Salmonella* spp. in cattle herds [13,16,20] but not in HROs. The model assessed the epidemiological impacts of *S*. Dublin on HROs by estimating cattle deaths and abortions. Findings related to the departure of asymptomatic and carrier individuals from HROs imply a potential role of these operations in the dissemination of the infection to dairy herds. Our model advances the understanding of *S*. Dublin transmission within a HRO by highlighting the important role of clinically ill cattle in herd-level spread, the reduced impact of carriers in *S*. Dublin dissemination past the initial pathogen introduction, and the relevance of pen environment contamination as a mechanism for infecting incoming calves. Furthermore, cleaning improvements and vaccination with a commercial vaccine were evaluated, providing insight into the expected epidemiological population-level impacts of these control strategies, if accessible to farmers. The economic module allows exploring the consequences of *S*. Dublin in HROs and determining under what conditions vaccination and cleaning improvements could help mitigate the economic impacts of these effects. This is relevant as the economic feasibility of implementing mitigation strategies against *S*. Dublin has not been explored in the US. In the US, approximately 12% of medium-sized farms (100–499 cows) and 44% of large dairy farms (>500 cows) raise heifers off-site [24], supporting the usefulness and relevance of our model. Key characteristics of most HROs in the US, such as the absence of pre-weaned and lactating cattle, limit direct application of our model or its findings to other cattle systems. Additionally, findings from our model should not be applied to operations that raise cattle in open lots or pastures, as the environmental conditions in those settings meaningfully differ from those in barns. Nonetheless, certain insights from the model, particularly the importance of environmental contamination in sustaining transmission, may still be useful for understanding *S*. Dublin dynamics in other intensively managed systems. For instance, while most feedlots differ from HROs in their open-lot design, some share management features such as the grouping of animals by age and structured cohort movement. These similarities suggest that the model's structure could be adapted to explore transmission dynamics in feedlot settings. Overall, we expect the model to be used by other researchers to predict the epidemiological and economic consequences of *S*. Dublin in HROs. As a mathematical abstraction describing *S*. Dublin spread within a herd of growing cattle operating on a batch system, general insights from our model could be of value to other cattle systems that could be similarly abstracted.

## HROs might be playing a relevant role in the dissemination of *S*. Dublin to other herds

Our findings imply that HROs can play a critical role in the transmission of *S*. Dublin to dairy farms through the movement of asymptomatic and carrier animals. This role may be particularly important in herds newly exposed to the pathogen, with ongoing transmission potential as the number of carriers increases over time. Consequently, these findings suggest that, unless effectively controlled within the HRO, such operations could serve as persistent sources of *S*. Dublin dissemination in both the short and long term. Importantly, carriers can become active shedders after experiencing stressful events, such as those during transportation [10], increasing the risk of *S*. Dublin spreading once heifers depart the HRO. The use of HROs by dairy farmers has been previously pointed out as a risk factor for introducing bacterial pathogens, including *Salmonella* [75]. Furthermore, Edrington et al. [76] indicated that 80% of fecal samples collected from heifers in a HRO were *Salmonella* positive right before returning to their facility of origin at 24 months old. Although our model predicted that the development of carriers during the first year could be reduced by more than half with stringent cleaning measures, the occurrence of asymptomatic individuals was less sensitive to this control strategy. This observation is likely due to the influx of susceptible individuals arriving in batches that are exposed to environmental contamination with the pathogen, allowing the infection to persist in the herd. These findings emphasize that HROs free of *S*. Dublin should prioritize preventing the introduction of the pathogen, as once established, even stringent cleaning measures may be insufficient to eradicate it and prevent its spread to other farms. Furthermore, since most infected individuals do not become carriers and remain infectious for approximately two weeks [14,28], these findings suggest that strategies such as pre-movement

testing and on-arrival quarantine could help prevent the introduction of *S*. Dublin to farms that use off-site heifer raising. A recent Danish study [77] found that cattle movements in the month preceding detection were the strongest predictor of *S*. Dublin infection in a herd, underscoring the importance of movement-related biosecurity. However, the effectiveness of pre-movement testing depends on the availability of reliable diagnostics, which remains a major limitation. Currently available tests have low sensitivity [11,48]. When within-herd prevalence is low, detecting infected herds becomes particularly challenging due to the reduced likelihood of sampling a positive animal [78]. Thus, while pre-movement testing holds promise in principle, significant improvements in diagnostic accuracy would be necessary before it could be implemented as a reliable surveillance strategy.

### Vaccination using a commercial vaccine reduces *S*. Dublin-induced mortality but slightly contributes to infection persistence in the herd

Reducing mortality resulting from *S*. Dublin infection allows farmers to have heifers ready to return to their farms of origin after the production period. However, findings from the model revealed possible adverse epidemiological consequences of vaccination based on the current knowledge of their effectiveness (Fig 3). Since available vaccination against *S*. Dublin has not been reported to prevent infection, nor shorten the duration of the infectious period, the infection continues to spread through the vaccinated herd (even better than in the absence of vaccination due to prevented deaths of infected individuals). Vaccinating infectious individuals to prevent them from dying eventually led to a slightly higher number of long-term carriers during the first and second years after the initial introduction of *S*. Dublin and the departure of a larger number of replacement heifers with an asymptomatic infection during the second year compared to baseline. The use of commercial vaccination also led to a reduction in the effectiveness of cleaning in preventing the development of carrier and asymptomatic individuals (S3 Table). A similar finding was observed by Lanzas et al. [18], who indicated that halving the probability of death after *Salmonella* infection in calves increased the median number of infected individuals. Overall, the silent spread of *S*. Dublin increases the level of infection in the herd, leading to the dissemination of the pathogen to other farms and the zoonotic risk to farm workers. Currently, the only confirmed effect of vaccination observed in practical settings is a decrease in the likelihood of *S*. Dublin-related death in young cattle after an extra-label administration [47], while a reduction in *S*. Dublin symptoms among vaccinated calves has been explored but lacks supporting evidence [46,47]. Nonetheless, our assessment of additional hypothetical vaccination scenarios indicates that even vaccines with low effectiveness (i.e., 25% reduction) in reducing susceptibility to *S*. Dublin, shortening the infectious period, and decreasing shedding can help lower *S*. Dublin-related mortality and abortions in HROs. These findings strongly suggest that such effects should be considered relevant during vaccine development and clinical trials.

### Stringent cleaning reduces *S*. Dublin outbreaks, helps prevent deaths and abortions, and limits the departure of infectious individuals to their facility of origin

Increasing the frequency of cleaning in the barn through multiple scrapings per day has important implications for the control of *S*. Dublin outbreaks in HROs. By continuously removing feces, this practice prevents cattle's contact with *S*. Dublin in the environment, thus helping prevent deaths, abortions, and the establishment of asymptomatic and carrier infections among replacement heifers. Additionally, more stringent cleaning reduces the likelihood that individuals are returned to their facility of origin being infectious, where they could contribute to further spread. These implications highlight the value of routine cleaning as a practical and effective component of *S*. Dublin management.

Previous studies have highlighted the importance of feces removal from the environment as a way to prevent the occurrence of secondary cases of *Salmonella* [13,16,20] and limit the duration of outbreaks [18]. In their study, Nielsen et al. [16] found that improving hygiene on a farm, understood as a reduction in the probability of infection from the environment, reduced by more than half the probability of an outbreak and meaningfully decreased the duration and size of

the epidemic. Similarly, Xiao et al. [13] found that frequently and effectively removing feces from the barn meaningfully contributed to controlling *Salmonella* infection by limiting its indirect transmission among cattle. Recent findings from a cross-sectional study done by Perry et al. [79] in Ontario dairy farms contradict predictions from our model by indicating that frequent manure removal from the calving area increases the risk of *S.* Dublin infection, suggesting that bacteria might spread across the barn during the manure removal process. However, Perry et al. [79] acknowledge that their findings might be a false positive related to farmers adopting frequent cleaning in the calving area after experiencing a *S.* Dublin outbreak. Nevertheless, the potential positive and negative impacts of floor scrapers on the spread of enteric bacteria in the barn need further investigation [80]. Unlike prior modeling efforts assessing hygiene as a control strategy, our approach provides a more realistic evaluation by accounting for the weekly or daily cleaning frequency of the barn's floor, representing observed or hypothetical cleaning practices on dairy farms. It is important to note that some of the stringent cleaning frequencies evaluated in this study, such as cleaning 6x or 12x times per day, may not be feasible in US HROs at present. However, our goal with this assessment was to determine the effectiveness of these practices if they were to be implemented in a HRO and whether developing the technology to achieve it would be a worthwhile resource investment. Further research is warranted to comprehensively understand the impact of cleaning on preventing *S.* Dublin outbreaks and reducing transmission. This should consider various management measures beyond increasing cleaning frequency, evaluated while considering animal welfare, environmental impacts (e.g., influence on ammonia emissions), and economic factors. Additionally, relying solely on cleaning is not sufficient to fully protect HROs from an outbreak and control the spread of the pathogen. Therefore, we suggest the adoption of a comprehensive approach to disease management that includes measures in addition to improvements in cleaning, such as limiting the number of farms providing heifers to the HRO, isolation and quarantine practices when heifers arrive at the operation, and efforts to early detect and remove *S.* Dublin-infected cattle.

### HROs experience a meaningful reduction in operating income following the introduction of *S.* Dublin, which can be partly prevented via stringent cleaning practices

*S.* Dublin poses important financial challenges for HROs due to factors such as treatment costs for severely affected heifers, losses from animals that die before departure, the opportunity cost of diverting heifers from their intended role as pregnant replacements to being sold for meat, and expenses related to rendering dead animals. Although the reduction in mortality due to vaccination increased the HRO's income by reducing heifer losses, in most scenarios this benefit was offset by the cost of administering the vaccine. It is important to note that the vaccine's effectiveness in improving the operating income may be more relevant in production settings that include newborns, such as calf raisers and some HROs, as they are more susceptible to experiencing severe symptoms and dying from the infection. It is also important to highlight that the Gram-negative modified-live nature of the assessed vaccines can lead to anaphylactic episodes in young cattle [5]. Although these anaphylactic events have been described as "low", "occasional", or "anecdotal" in the literature [5,46,81], communications with dairy producers and veterinarians describe these events as a serious concern (as indicated by E.F.). The rate at which these anaphylactic events occur among individuals of different ages remains to be systematically evaluated, as well as its economic relevance for heifer raisers. Daily cleaning may offer a more economically viable and effective approach to limiting the impact of *S.* Dublin in HROs compared to the sole implementation of vaccination (with a vaccine that solely reduces mortality) and cleaning on a per week basis. This is the result of preventing mortality and, in contrast to vaccination, reducing the occurrence of severe cases and the overall level of infection in the farm through the frequent removal of feces. Importantly, stringent cleaning routines also reduce the likelihood of *S.* Dublin outbreaks altogether and thereby their associated financial losses.

Our findings from the economic assessment of cleaning improvements conflict with the assessment by Bergevoet et al. [82], who characterized hygienic measures as the least cost-effective approach for controlling *Salmonella* (serovars Typhimurium and Dublin) compared to other strategies such as segregating cattle by age groups and preventing the

introduction of new animals, although in that study it still reduced prevalence by half over a period of three years. However, it is important to note that Bergevoet et al. [82] assessed cleaning measures solely in terms of reducing the probability of developing chronic infection (i.e., carrier state), without considering the broader significance of indirect transmission in *Salmonella* dynamics. On the other hand, Nielsen et al. [32] found that dairy farms applying very good management practices, including excellent hygiene, had the highest improvements in profit. The improvement in the operating income observed when cleaning on a per day basis, particularly in scenarios with lower costs and frequent cleaning, suggests that innovations allowing farmers to clean cheaper and better can meaningfully improve a HRO's profitability.

We acknowledge that our assessment of cleaning frequency did not consider the potential negative effects on cattle behavior and comfort, which could lead to negative financial consequences. For example, Cramer et al. [83] and Crossley et al. [84] have found that increased cleaning frequency is associated with a higher occurrence of tail and hoof lesions in dairy cattle. While increased cleaning frequency may help mitigate *S*. Dublin, particularly alongside other measures, a better understanding of the effects of frequent cleaning (e.g., cleaning with an alley scraper multiple times per day) on cattle productivity and well-being would improve economic estimations. Our findings provide valuable insights into *S*. Dublin impacts on HROs in the US, filling an important knowledge gap since previous studies on economic losses associated with *S*. Dublin in dairy farms have been largely limited to European countries [32,82,85,86].

### Carriers are relevant for the initial introduction of *S*. Dublin into HROs but have minimal impact on transmission dynamics once the infection reaches an endemic state

Carriers appear to play an important role in the initial introduction and early spread of *S*. Dublin within HROs, though their influence on transmission after the infection becomes established seems limited. Instead, the persistence of *S*. Dublin over time is driven primarily by clinical cases within the herd and the continuous introduction of susceptible animals. This interpretation aligns with the study by Lanzas et al. [17], which showed that long-term shedders and subclinical individuals with lower *Salmonella* shedding had a minimal impact on the occurrence of infection cases in a dairy farm. The findings of Nielsen et al. [16] challenge this notion by highlighting the essential role played by carriers in the long-term persistence of *S*. Dublin in dairy herds. The disparity in carrier importance between their model and ours could stem from their consideration of increased resistance to reinfection in previously infected individuals, highlighting the significance of persistent shedders in sustaining the infection within a growing population of individuals with reduced susceptibility. Additionally, the HROs have a faster turnover rate compared to dairy farms (animals typically spend ~1.3 years in a HRO, whereas they spend ~5 years in a dairy farm [24,87], which might have also contributed to the difference in the relative importance of long-term shedders in sustaining the disease within these operations.

The limited role of carriers in spreading *S*. Dublin within an HRO suggests that a testing-and-culling approach targeting carriers is likely ineffective for controlling the infection. Despite efforts to identify and eliminate persistent shedders in a high-risk operation, incoming cattle may still become infected and develop a carrier state due to exposure to *S*. Dublin from other infectious cattle. Furthermore, the intermittent shedding of *S*. Dublin by carriers and the limitations of testing methods in detecting ongoing infections make it difficult to control this pathogen in HROs through testing-and-culling strategies [10,88]. As a result, our model supports previous findings that targeting carriers alone is unlikely to be a cost-effective strategy for managing *S*. Dublin in HROs.

### Identified knowledge gaps in *S*. Dublin epidemiology

The duration of immunity to *S*. Dublin infection ($D_R$) was identified as an important parameter determining model outcomes. This has also been reported by other mathematical modeling studies [13,17,20]. Unfortunately, the duration of the period included in our model is based on the humoral response of cattle [28,31], which does not necessarily indicate protection against *S*. Dublin reinfection. Indeed, the cellular response is the primary means of protection against *Salmonella* infection, with antibodies playing a secondary role [11].

In our study, we did not consider super shedders (i.e., individuals that shed into the environment a much larger amount of *S.* Dublin than other infectious individuals and for an extended period) in the transmission of *S.* Dublin due to its importance still being under discussion. For instance, Lanzas et al. [17] highlighted super shedders as important for transmitting *Salmonella* in dairies, with its occurrence leading to an increase in the number of secondary infections. In contrast, Nielsen et al. [16] indicated that super shedders are not an essential component in *S.* Dublin epidemiology, with their occurrence being extremely rare. In both studies, super shedders were modeled as infectious individuals shedding the same amount of *Salmonella* into the environment as clinically infected cattle but for an extended period. While our model did not explicitly consider super shedders, scenario analysis evaluated the impact of carrier individuals shedding the same amount of *S.* Dublin as clinically ill cattle (S3 Table). The results revealed a moderate difference in epidemiological outcomes, supporting that the super shedders could have a meaningful impact on *S.* Dublin transmission dynamics.

Another important knowledge gap is the dynamics of intermittent shedding, i.e., the frequency with which carriers interrupt and restart their shedding of *S.* Dublin. We chose not to incorporate intermittent shedding into the model because there is limited empirical data on how often and for how long carriers stop and resume shedding. Accounting for this variability would require introducing additional assumptions and parameters that are not well supported by current evidence. Moreover, when multiple animals are infected at the same time, intermittent shedding is unlikely to affect overall environmental contamination unless their shedding patterns are highly synchronized, which is improbable. Given these considerations, we opted for a simpler approach that assumes continuous shedding during the carrier state. Nonetheless, elucidating *S.* Dublin shedding intermittency would allow researchers to more precisely assess the role of carriers in *S.* Dublin dynamics in dairy cattle operations, since considering uninterrupted shedding (as we did in our model) may under- or overestimate their role as disseminators.

It is crucial to acknowledge the lack of information regarding the costs associated with implementing cleaning improvements and vaccination. Additional information is needed on the costs of using a skid steer or alley scraper at varying frequencies (e.g., per day or week) and across operations of different sizes to accurately determine their influence on the operating income of an HRO. Considering the benefits of cleaning improvements observed in our study, developing new technologies that enable frequent floor scraping at a lower cost could meaningfully improve the HRO's operating income in the face of a *S.* Dublin outbreak. For vaccination, understanding its economic implications is essential, particularly regarding labor-related costs for its initial administration and revaccination, as well as potential side effects in young cattle. Addressing these knowledge gaps will be key to developing economically sound strategies for controlling *S.* Dublin transmission.

## Model limitations

Because data suitable for model validation from a HRO were not available, we validated the model using information from a dairy farm that experienced a *S.* Dublin outbreak after the introduction of cattle in batches from another dairy farm [67]. While not ideal, this was considered acceptable because of the similarities between *S.* Dublin epidemiology on dairy farms and HROs. As more *S.* Dublin data on HROs become available, the model parameters and validation analysis can be updated. The model did not account for increased resistance to *S.* Dublin infection among susceptible individuals who had previously been infected and returned to the susceptible compartment upon the loss of immunity [89]. This exclusion was due to the limited data for parameterization. The inclusion of this feature in the model could have reduced the incidence of *S.* Dublin reinfections in previously exposed cattle and ultimately decreased health impacts. However, our findings are relevant to a naive herd and thus represent conservative predictions. Subject to data availability, future modeling efforts should consider resistance to reinfection when emulating *S.* Dublin dynamics in cattle. The homogeneous mixing of cattle in a pen was chosen as a simplification for model parameterization, a choice also made by other *Salmonella* modeling studies [18,20,21]. The assumption of homogeneous mixing considered in our model contributes to increasing the chances of a single individual causing an outbreak and spreading disease. Despite this limitation, the validation process done to the model indicates that it is capable of predicting the level of infection observed in cattle farms, therefore

although a simplification, it provides a valuable approximation to the situation in real life. Due to the compartmental nature of the model, vaccination was incorporated at the start of the simulation, rather than in response to an outbreak. Furthermore, our model focused on within-farm transmission, assuming that indirect contact through contaminated environments is the primary route of infection among cattle. As a result, the broader implications of animal movement between farms, such as the risk of introducing infection to naïve herds during transportation and triggering outbreaks, were beyond the scope of this study and therefore not assessed. We acknowledge this as a limitation of our study, since farm-to-farm transmission through cattle movement has been identified as an important mechanism for *S.* Dublin spread between dairy operations [77]. Future studies evaluating transmission between HROs and farms would provide valuable insights into the role played by these operations in the broader dissemination of *S.* Dublin in the US. The limitations of the model also prevented considering a two-week interval between vaccine doses, most likely leading to a slight overestimation of its effectiveness in preventing deaths among calves and young heifers. Additionally, we only evaluated scenarios where mitigation strategies, such as cleaning improvements and vaccination, were implemented simultaneously with the introduction of *S.* Dublin into a naïve herd. We did not assess these interventions implemented as preventative measures during times when the introduction of the infection into the herd is uncertain (not occurring at all or occurring with a low probability). The epidemiological and economic outcomes of these interventions would obviously be different in such scenarios compared to the results presented here. Analysis of such uncertainty scenarios was not conducted because it would require assumptions about the timing and scale of *S.* Dublin introduction events and the history of infection and immunity in the herd. While this presents a limitation, we believe the scenarios analyzed here are informative, considering that HROs receive animals from multiple dairy herds, increasing the risk of infection introduction with the most severe consequences for a naïve herd as assumed here.

## Conclusions

The developed mathematical model of *S.* Dublin transmission dynamics in a HRO predicts the epidemiological and economic consequences of *S.* Dublin infection in a US HRO over a 2-year simulation period, offering valuable insights for disease management. Our findings imply that HROs play in the dissemination of *S.* Dublin to other herds through the movement of asymptomatic and persistently infected individuals. Additionally, the model reveals that, although carriers are important in the initial introduction of *S.* Dublin into HROs, the short production period for heifers in HRO (less than two years) prevents carriers from meaningfully contributing to the persistence of the infection in the herd. Among evaluated control strategies, stringent cleaning practices substantially mitigate *S.* Dublin impacts on a HRO, preventing deaths and abortions, and limiting the departure of infectious heifers to their facility of origin. Meanwhile, vaccination with a hypothetical vaccine, capturing current knowledge of the effectiveness of commercial *S.* Dublin vaccines, reduces *S.* Dublin-induced mortality but also slightly contributes to infection persistence within the herd. Thus, based on the current information about these vaccines, alternative measures to vaccination are necessary to sustainably control *S.* Dublin on HROs. This also calls for more research into vaccines against *S.* Dublin infection. The study also demonstrated that *S.* Dublin introduction leads to a meaningful reduction in a HRO's operating income, a loss that can be partly offset by implementing frequent cleaning. Knowledge gaps, including the cost of interventions, intermittent shedding dynamics, and the duration of immunity, remain barriers to fully understanding *S.* Dublin dynamics on HROs. Addressing these gaps is essential for designing more effective control measures against the pathogen and improving economic outcomes for HROs. Overall, our findings support the importance of addressing *S.* Dublin as a notable threat to dairy cattle production in the US and suggests the need for control at HROs to prevent its potential dissemination to dairy farms.

## Supporting information

**S1 Appendix. Details about the methodology and description of additional findings.**
(DOCX)

**S2 Appendix. Tool sheet used to determine cleaning effectiveness per day at different cleaning frequencies in a heifer-raising operation.** Information about cleaning effectiveness was obtained through expert elicitation.
(XLSX)

**S1 Table. Comparison of predictions obtained from the *Salmonella* Dublin model at the end of a 2-year long simulation period with 100, 500, 1,000, and 5,000 iterations.** The number of cattle on the farm (N) is 1,000.
(DOCX)

**S2 Table. Summary of scenarios assessed to determine the probability of an outbreak (PoO), epidemiological outcomes (DACA: deaths, abortions, and carriers and asymptomatic infections among raised replacement heifers), and/or the operating income (OI) in a heifer-raising operation (HRO).** Cells marked with an 'X' indicate that the corresponding outcome was estimated in that scenario.
(DOCX)

**S3 Table. Detailed epidemiological findings from the scenario analyses evaluated using the developed *Salmonella* Dublin model.** The comparison was made based on the probability of an outbreak and the number of deaths, abortions, carrier replacement heifers, and asymptomatic replacement heifers by the end of the first and second years into the simulation. This table corresponds to the results shown in Fig 7 in the main text. Scenarios are described in S2 Table.
(DOCX)

**S4 Table. Operating income (USD per 100-head) observed in scenarios with different costs required for the implementation of mitigation strategies.** A comparison in terms of the operating income (USD per 100-head) was made between the baseline scenario (cleaning once (1x) per week and no vaccination) and scenarios in which control strategies were implemented. These strategies included cleaning at different frequencies (3x, 5x, and 7x times per week and 2x, 4x, 6x, 8x, and 12 per day) and vaccination reducing *S.* Dublin-related mortality (vaccine effectiveness: median = 0.72, 5th-95th percentile = 0.48–0.91). Operating income in a "no infection" scenario = USD 39,123 per 100-head per year. This table corresponds to the results shown in Fig 8 in the main text.
(DOCX)

**S1 Fig. *Salmonella* Dublin dynamics in a heifer-raising operation with (top) and without (bottom) considering natural mortality among cattle.**
(TIFF)

**S2 Fig. Operating income (USD per 100-head) observed in scenarios with different costs required for the implementation of mitigation strategies.** A comparison in terms of the operating income (USD per 100-head) was made between the baseline scenario (cleaning once (1x) per week and no vaccination) and scenarios in which multiple control strategies were implemented. Observations for the first and second years into the simulation are presented in the figure. Horizontal grey lines in each plot indicate the median value of the corresponding outcome under the baseline scenario.
(TIF)

## Acknowledgments

The authors would like to extend their appreciation to Dr. Angel Abuelo, Dr. Pablo Pinedo, Dr. Kaitlyn Kremer, Dr. Erin Goodrich, Dr. Eleni Casseri, and Dr. Trent Westhoff for their insights about cleaning effectiveness and vaccination. We very much appreciate the assistance of Dr. Peter Thomsen for the valuable information that he provided about Danish dairy farms.

## Author contributions

**Conceptualization:** Renata Ivanek.

**Formal analysis:** Sebastian Llanos-Soto.

**Funding acquisition:** Renata Ivanek.

**Investigation:** Sebastian Llanos-Soto, Renata Ivanek.

**Methodology:** Sebastian Llanos-Soto, Martin Wiedmann, Aaron Adalja, Christopher Henry, Paolo Moroni, Elisha Frye, Francisco A. Leal Yepes, Renata Ivanek.

**Project administration:** Sebastian Llanos-Soto.

**Resources:** Renata Ivanek.

**Supervision:** Renata Ivanek.

**Validation:** Sebastian Llanos-Soto.

**Visualization:** Sebastian Llanos-Soto.

**Writing – original draft:** Sebastian Llanos-Soto.

**Writing – review & editing:** Sebastian Llanos-Soto, Martin Wiedmann, Paolo Moroni, Elisha Frye, Renata Ivanek.

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
