## [Decision Letter · Decision Letter 0]

10 Mar 2025

PONE-D-25-05247Integration of mathematical modeling and economics approaches to evaluate strategies for control of Salmonella Dublin in a heifer-raising operationPLOS ONE

Dear Dr. Llanos-Soto,

Thank you for submitting your manuscript to PLOS ONE. After careful consideration, we feel that it has merit but does not fully meet PLOS ONE’s publication criteria as it currently stands. Therefore, we invite you to submit a revised version of the manuscript that addresses the points raised during the review process. Both reviewers have made some great suggestions for this manuscript. Please consider them carefully. As noted by one of the reviewers, the manuscript tries to cover a lot of ground, which is great but also sometimes difficult to manage. As noted, splitting the manuscript into 2 articles is a possibility, but I would encourage the authors to keep everything condensed in one as is, for which some additional details and re-structuring is warranted, as suggested below.

We look forward to receiving your revised manuscript.

Kind regards,

Angel Abuelo, DVM, MRes, MSc, PhD, DABVP (Dairy), DECBHM

Academic Editor

PLOS ONE

Journal Requirements:

“This material is based upon work supported by the National Institute of Food and Agriculture, USDA (Washington, DC), Hatch, under accession number 7000433, as well as Multistate Research Funds, accession numbers 1016738 and 7005699 awarded to R. I. This project was also supported by the Cornell Institute of Digital Agriculture (CIDA) through the Research  Innovation Fund (RIF) awarded to SL-S.”

“R.I. was funded by the National Institute of Food and Agriculture, USDA (Washington, DC), Hatch, under accession number 7000433 and Multistate Research Funds, accession numbers 1016738 and 7005699. URL: https://www.nifa.usda.gov/

S. L.-S. was supported by the Cornell Institute of Digital Agriculture (CIDA) through the Research Innovation Fund (RIF). URL: https://digitalagriculture.cornell.edu/research-support/.

Funding sources did not play any role in the study design, data collection and analysis, decision to publish, or preparation of the manuscript.”

Reviewers' comments:

Reviewer's Responses to Questions

**Comments to the Author**

1. Is the manuscript technically sound, and do the data support the conclusions?

Reviewer #1: Partly

Reviewer #2: Yes

2. Has the statistical analysis been performed appropriately and rigorously? 

Reviewer #1: Yes

Reviewer #2: Yes

3. Have the authors made all data underlying the findings in their manuscript fully available?

Reviewer #1: Yes

Reviewer #2: Yes

4. Is the manuscript presented in an intelligible fashion and written in standard English?

Reviewer #1: Yes

Reviewer #2: Yes

5. Review Comments to the Author

Reviewer #1: Dear Author's,

Thank you for the opportunity to review your manuscript. Overall, the paper is well written, and the mathematical model is clear and easy to follow from both an epidemiological and economic perspective. I appreciate the depth of analysis and the valuable insights provided. Below are a few comments that may help strengthen the study:

1. Benefit of the Study to Modeled HRO

It would be helpful to explicitly highlight how this study enhances the understanding of S. Dublin transmission within a single heifer-raising operation (HRO).

Additionally, since the model is based on an HRO, discussing its applicability to other cattle herd types would add value.

2. Realism of Cleaning Assumption

The assumption of cleaning a farm 12 times per day seems quite intensive. The authors could discuss the realism of this scenario and, if possible, provide references or practical justifications for this assumption.

3. Transmission Pathways and Indirect Transmission Assumption

The study assumes that susceptible individuals become infected primarily through indirect transmission via the environment contaminated with S. Dublin bacteria. However, some studies suggest that farm-to-farm movements are the main driver of spread (e.g., https://www.sciencedirect.com/science/article/pii/S0022030224008166).

Since the authors acknowledge in line 526 that HROs play a role in dissemination, and again in line 778 regarding cattle movements, it would be useful to discuss S. Dublin spread via movements in more detail.

4. Testing and Quarantine Strategies

Line 547 mentions that testing cattle before returning to their farms and quarantining animals upon arrival could help prevent the introduction of S. Dublin.

It would be helpful to discuss how long before returning to the farm testing should occur. The authors may refer to the Danish study mentioned above (https://www.sciencedirect.com/science/article/pii/S0022030224008166) for additional context.

5. Test Sensitivity and Within-Herd Prevalence

Line 551 states that the available tests have low sensitivity. The authors could briefly discuss how test sensitivity varies with within-herd prevalence, as this could impact the effectiveness of detection and control strategies.

6. Duration of the Simulation Period

The manuscript simulates S. Dublin spread over a two-year period (lines 275 and 777).

Given that the Danish study mentioned above reports a median infection duration of 25 months in farms, which aligns well with the chosen simulation period, the authors could briefly discuss how model results would be influenced by choosing a longer or shorter period.

I appreciate the effort that went into this study and believe these additions will further enhance its clarity and impact. I look forward to seeing the revised version.

Reviewer #2: This manuscript presents a stochastic simulation model to assess the impact of Salmonella Dublin infection and evaluate the cost-effectiveness of mitigation strategies in heifer-raising operations (HROs). The integration of an economic analysis with a modified Susceptible-Infected-Recovered-Susceptible (SIRS) model is a major strength, as it provides a novel and practical approach to understanding S. Dublin transmission dynamics and control measures. Given the increasing concern surrounding S. Dublin in dairy production and its public health implications, this study offers valuable insights for stakeholders, including farmers, policymakers, and researchers. The inclusion of model code in a public repository (GitHub) further enhances the study's transparency and reproducibility, which is commendable.

Overall, this study presents a strong model with practical implications, but refining its focus and clarity will maximize its utility for both researchers and industry stakeholders. I believe the study currently attempts to cover too much ground in a single manuscript and would be beneficial to split it into two papers: one focusing on the within-farm model itself (per se already innovative), including sensitivity analyses of key parameters, and another dedicated to the economic evaluation of mitigation strategies. If the authors prefer to keep the structure as is, then some details need to be made clearer in the text, and the inclusion of some supplementary information in the core text of the manuscript should be reconsidered.

6. PLOS authors have the option to publish the peer review history of their article (what does this mean? ). If published, this will include your full peer review and any attached files.

**Do you want your identity to be public for this peer review?** For information about this choice, including consent withdrawal, please see our Privacy Policy .

Reviewer #1: No

Reviewer #2: No

---

## [Author Response · Author response to Decision Letter 1]

17 Aug 2025

Dear Dr. Abuelo,

Thank you very much for your comments. We made the format and file name changes to supplementary figures and tables that were requested in the decision letter. We would also appreciate if the "Funding statement" could be updated as indicated in our cover letter.

The new version of the manuscript addresses reviewers' suggestions, including:

- Expanding our justification for developing a model specific to S. Dublin, emphasizing its distinct epidemiology and relevance to public health.

- Highlighting our model’s contribution to understanding transmission within heifer-raising operations, its limitations, and generalizability to other systems.

- Revising the manuscript’s structure, avoiding redundancy, and improving clarity. This includes moving sections from “S1_Appendix” into the main text to provide all the necessary information for understanding the study.

- Clarifying model assumptions regarding cleaning frequency, direct transmission, and intermittent shedding.

- Incorporating additional scenario analyses exploring hypothetical vaccine effects.

Our complete and detailed response to reviewers can be found in a friendly-to-read format in the "Response to Reviewers" document uploaded during the revising process. These are our responses to the comments indicated specifically in the Decision letter (which are also addressed in our Response to Reviewers" document:

Reviewer #1:

Comment: Benefit of the Study to Modeled HRO: It would be helpful to explicitly highlight how this study enhances the understanding of S. Dublin transmission within a single heifer-raising operation (HRO). Additionally, since the model is based on an HRO, discussing its applicability to other cattle herd types would add value.

R. Thank you very much for your comment. We included additional sentences in the “Discussion” section to indicate the contribution of our model to understanding S. Dublin transmission within a HRO. Specifically, we now indicate the following in the discussion section:

“Our model advances the understanding of S. Dublin transmission within a HRO by highlighting the important role of clinically ill cattle in herd-level spread, the reduced impact of carriers in S. Dublin dissemination past the initial pathogen introduction, and the relevance of pen environment contamination as a mechanism for infecting incoming calves” (Lines 638-642).

Regarding the generalizability of our model to other systems, we included sentences to indicate that the extent to which our model applies to other cattle operations is restricted (Lines 650-664):

“Key characteristics of most HROs in the US, such as the absence of pre-weaned and lactating cattle, limit direct application of our model or its findings to other cattle systems. Additionally, findings from our model should not be applied to operations that raise cattle in open lots or pastures, as the environmental conditions in those settings meaningfully differ from those in barns. Nonetheless, certain insights from the model, particularly the importance of environmental contamination in sustaining transmission, may still be useful for understanding S. Dublin dynamics in other intensively managed systems. For instance, while most feedlots differ from HROs in their open-lot design, some share management features such as the grouping of animals by age and structured cohort movement. These similarities suggest that the model’s structure could be adapted to explore transmission dynamics in feedlot settings. Overall, we expect the model to be used by other researchers to predict the epidemiological and economic consequences of S. Dublin in HROs. As a mathematical abstraction describing S. Dublin spread within a herd of growing cattle operating on a batch system, general insights from our model could be of value to other cattle systems that could be similarly abstracted.”

Comment: Realism of Cleaning Assumption: The assumption of cleaning a farm 12 times per day seems quite intensive. The authors could discuss the realism of this scenario and, if possible, provide references or practical justifications for this assumption.

R. We understand your concern about frequent cleaning. The intervention scenarios involving frequent cleaning and vaccination were designed not only to reflect current practices but also to explore potential approaches that are currently impractical/unfeasible and would require further technological development for their implementation. For instance, if our study findings had supported the notion that frequent cleaning is ineffective, there would be little justification for investing in new technologies to support it. Additionally, if more effective cleaning becomes practically feasible, it would also offer additional benefits, such as reducing ammonia emissions. We updated the “Discussion” (Lines 754–758 and lines 761–762) section to highlight potential additional benefits of developing strategies for improved cleaning and vaccination. The edited text reads as follows:

“It is important to note that some of the stringent cleaning frequencies evaluated in this study, such as cleaning 6x or 12x times per day, may not be feasible in US HROs at present. However, our goal with this assessment was to determine the effectiveness of these practices if they were to be implemented in a HRO and whether developing the technology to achieve it would be a worthwhile resource investment.”

Comment: Transmission Pathways and Indirect: Transmission Assumption The study assumes that susceptible individuals become infected primarily through indirect transmission via the environment contaminated with S. Dublin bacteria. However, some studies suggest that farm-to-farm movements are the main driver of spread (e.g., https://www.sciencedirect.com/science/article/pii/S0022030224008166). Since the authors acknowledge in line 526 that HROs play a role in dissemination, and again in line 778 regarding cattle movements, it would be useful to discuss S. Dublin spread via movements in more detail.

R. Thank you for your comment. We agree that farm-to-farm movements, particularly through the movement of animals, can play an important role in the dissemination of S. Dublin between dairy operations. We now acknowledge this in the “Discussion”, specifically in the “Model limitation” section (Lines 912–921). Briefly, we now indicate that our model does not account for the transmission of S. Dublin between farms via animal movements, which has been found to play an important role in S. Dublin spread elsewhere. The paragraph reads as follows:

“Furthermore, our model focused on within-farm transmission, assuming that indirect contact through contaminated environments is the primary route of infection among cattle. As a result, the broader implications of animal movement between farms, such as the risk of introducing infection to naïve herds during transportation and triggering outbreaks, were beyond the scope of this study and therefore not assessed. We acknowledge this as a limitation of our study, since farm-to-farm transmission through cattle movement has been identified as an important mechanism for S. Dublin spread between dairy operations [77]. Future studies evaluating transmission between HROs and farms would provide valuable insights into the role played by these operations in the broader dissemination of S. Dublin in the US.”

Comment: Testing and Quarantine Strategies: Line 547 mentions that testing cattle before returning to their farms and quarantining animals upon arrival could help prevent the introduction of S. Dublin. It would be helpful to discuss how long before returning to the farm testing should occur. The authors may refer to the Danish study mentioned above (https://www.sciencedirect.com/science/article/pii/S0022030224008166) for additional context

R. Thank you very much for this comment. We incorporated the suggested article into the discussion to highlight the importance of the timing of testing to prevent the dissemination of S. Dublin to other operations (Lines 689–694). Nonetheless, we indicate that testing has severe limitations at the moment and improvements are needed to implement it as a surveillance strategy (Lines 694–697). The paragraph reads as follows:

“A recent Danish study [77] found that cattle movements in the month preceding detection were the strongest predictor of S. Dublin infection in a herd, underscoring the importance of movement-related biosecurity. However, the effectiveness of pre-movement testing depends on the availability of reliable diagnostics, which remains a major limitation. Currently available tests have low sensitivity [11,48]. When within-herd prevalence is low, detecting infected herds becomes particularly challenging due to the reduced likelihood of sampling a positive animal [78]. Thus, while pre-movement testing holds promise in principle, significant improvements in diagnostic accuracy would be necessary before it could be implemented as a reliable surveillance strategy.”

Comment: Test Sensitivity and Within-Herd Prevalence: Line 551 states that the available tests have low sensitivity. The authors could briefly discuss how test sensitivity varies with within-herd prevalence, as this could impact the effectiveness of detection and control strategies

R. Thanks for the suggestion. We now include an additional sentence to indicate the particular importance of test sensitivity in scenarios with low within-herd prevalence (Lines 694–697). The updated sentences indicate the following:

“When within-herd prevalence is low, detecting infected herds becomes particularly challenging due to the reduced likelihood of sampling a positive animal [78]. Thus, while pre-movement testing holds promise in principle, significant improvements in diagnostic accuracy would be necessary before it could be implemented as a reliable surveillance strategy.”

Comment: Duration of the Simulation Period: The manuscript simulates S. Dublin spread over a two-year period (lines 275 and 777). Given that the Danish study mentioned above reports a median infection duration of 25 months in farms, which aligns well with the chosen simulation period, the authors could briefly discuss how model results would be influenced by choosing a longer or shorter period

R. Thanks for your comment. We now include clarification in the methods to justify the decision to select 2-years as the simulation period (Lines 347–350):

“The 2-year simulation length was selected because S. Dublin dynamics show no remarkable variations after two years into the simulation, while selecting a shorter period provides a narrower understanding of S. Dublin epidemiological and economic impacts on a HRO.”

Reviewer #2

Comment: This manuscript presents a stochastic simulation model to assess the impact of Salmonella Dublin infection and evaluate the cost-effectiveness of mitigation strategies in heifer-raising operations (HROs). The integration of an economic analysis with a modified Susceptible-Infected-Recovered-Susceptible (SIRS) model is a major strength, as it provides a novel and practical approach to understanding S. Dublin transmission dynamics and control measures. Given the increasing concern surrounding S. Dublin in dairy production and its public health implications, this study offers valuable insights for stakeholders, including farmers, policymakers, and researchers. The inclusion of model code in a public repository (GitHub) further enhances the study's transparency and reproducibility, which is commendable.

Overall, this study presents a strong model with practical implications, but refining its focus and clarity will maximize its utility for both researchers and industry stakeholders. I believe the study currently attempts to cover too much ground in a single manuscript and would be beneficial to split it into two papers: one focusing on the within-farm model itself (per se already innovative), including sensitivity analyses of key parameters, and another dedicated to the economic evaluation of mitigation strategies. If the authors prefer to keep the structure as is, then some details need to be made clearer in the text, and the inclusion of some supplementary information in the core text of the manuscript should be reconsidered.

R. We sincerely thank the reviewer for their positive assessment of our work and for highlighting the novelty and practical relevance of integrating an economic analysis with a modified SIRS model. We appreciate the recognition of the study’s value to stakeholders and the emphasis on transparency through sharing model code. Regarding the suggestion to split the manuscript into two separate papers, after careful consideration we decided to maintain the current structure, while implementing the reviewer’s recommendation to refine focus and clarity. Specifically, we moved key details from the supplementary materials (including model equations) into the main text, removed repetitive information, and added clarifications where needed to improve readability. We believe these revisions address the reviewer’s concern while preserving the comprehensive perspective of both the epidemiological and economic aspects of the model within a single manuscript. For more information on manuscript changes, please check the "Response to Reviewers" document.

We deeply appreciate the editor and reviewers' feedback during this process.

Sincerely on behalf of the co-authors,

Sebastian Llanos-Soto

---

## [Decision Letter · Decision Letter 1]

1 Sep 2025

Integration of mathematical modeling and economics approaches to evaluate strategies for control of Salmonella Dublin in a heifer-raising operation

PONE-D-25-05247R1

Dear Dr. Llanos-Soto,

We’re pleased to inform you that your manuscript has been judged scientifically suitable for publication and will be formally accepted for publication once it meets all outstanding technical requirements.

Kind regards,

Angel Abuelo, DVM, MRes, MSc, PhD, DABVP (Dairy), DECBHM

Academic Editor

PLOS ONE

Additional Editor Comments (optional):

Reviewer #1:

Reviewer #2:

Reviewers' comments:

Reviewer's Responses to Questions

**Comments to the Author**

1. If the authors have adequately addressed your comments raised in a previous round of review and you feel that this manuscript is now acceptable for publication, you may indicate that here to bypass the “Comments to the Author” section, enter your conflict of interest statement in the “Confidential to Editor” section, and submit your "Accept" recommendation.

Reviewer #1: All comments have been addressed

Reviewer #2: All comments have been addressed

2. Is the manuscript technically sound, and do the data support the conclusions?

Reviewer #1: Yes

Reviewer #2: Yes

3. Has the statistical analysis been performed appropriately and rigorously? 

Reviewer #1: Yes

Reviewer #2: Yes

4. Have the authors made all data underlying the findings in their manuscript fully available?

Reviewer #1: Yes

Reviewer #2: Yes

5. Is the manuscript presented in an intelligible fashion and written in standard English?

Reviewer #1: Yes

Reviewer #2: Yes

6. Review Comments to the Author

Reviewer #1: The authors have thoroughly addressed my comments and incorporated the suggested revisions appropriately. They have also provided clear and satisfactory explanations where changes were not required. I am satisfied with their responses and consider the manuscript suitable for acceptance.

Reviewer #2: We appreciate the author’s effort in addressing every single comment. I believe all revisions greatly strengthened the work, and we recognize the merit of the research. PLOS ONE publication criteria is met: the research is original, analyses are performed to a high technical standard and clearly described, conclusions are well supported by the data, and the manuscript is well written.

Minor note for consideration:

Lines 161–167: Thank you for clarifying the transmission routes of S. Dublin. We suggest simplifying the text to:

“We have considered fecal-oral transmission via the shared environment as the exclusive transmission route in our model [10], given that direct transmission through recto-anal ingestion of fecal material and respiratory route [5] is believed less common.”, or something similar in short.

Thank you again for politely addressing my comments and best of luck with your research!

7. PLOS authors have the option to publish the peer review history of their article (what does this mean? ). If published, this will include your full peer review and any attached files.

**Do you want your identity to be public for this peer review?** For information about this choice, including consent withdrawal, please see our Privacy Policy .

Reviewer #1: No

Reviewer #2: No

---

## [Editor Report · Acceptance letter]

PONE-D-25-05247R1

PLOS ONE

Dear Dr. Llanos-Soto,

I'm pleased to inform you that your manuscript has been deemed suitable for publication in PLOS ONE. Congratulations! Your manuscript is now being handed over to our production team.

Kind regards,

on behalf of

Dr. Angel Abuelo

Academic Editor

PLOS ONE